# AoPS Dataset: Leveraging Online Olympiad-Level Math Problems for LLMs Training and Contamination-Resistant Evaluation

## Abstract

Advances in Large Language Models (LLMs) have sparked interest in their ability to solve Olympiad-level math problems. However, the training and evaluation of these models are constrained by the limited size and quality of available datasets, as creating large-scale data for such advanced problems requires extensive effort from human experts. In addition, current benchmarks are prone to contamination, leading to unreliable evaluations. In this paper, we present an automated pipeline that leverages the rich resources of the Art of Problem Solving (AoPS) forum, which predominantly features Olympiad-level problems and community-driven solutions. Using open-source LLMs, we develop a method to extract question-answer pairs from the forum, resulting in **AoPS-Instruct**, a dataset of more than 650,000 high-quality QA pairs. Our experiments demonstrate that fine-tuning LLMs on AoPS-Instruct improves their reasoning abilities across various benchmarks. Moreover, we build an automatic pipeline that introduces **LiveAoPSBench**, an evolving evaluation set with timestamps, derived from the latest forum data, providing a contamination-resistant benchmark for assessing LLM performance. Notably, we observe a significant decline in LLM performance over time, suggesting their success on older examples may stem from pre-training exposure rather than true reasoning ability. Our work presents a scalable approach to creating and maintaining large-scale, high-quality datasets for advanced math reasoning, offering valuable insights into the capabilities and limitations of LLMs in this domain. Our benchmark is available at https://livemathbench.github.io/leaderboard.

## 1 Introduction

Large language models (LLMs) have shown tremendous success in solving various tasks such as code generation (Li et al., 2022), math reasoning (Shao et al., 2024), and commonsense reasoning (Zellers et al., 2019; Achiam et al., 2023), suggesting that current models may show signs of artificial general intelligence (AGI) (Bubeck et al., 2023). Math reasoning is perhaps one of the most challenging tasks for the LLMs, since mathematics is inherently structured, requiring not just recall of facts but also rigorous logical inference, abstraction, and understanding of formal symbolic systems. As such, there have been grand challenges (Selsam et al., 2019) and million-dollar prizes AIMO (2023) established for a model capable of solving Olympiad-level math problems.

On the training side, despite significant progress in certain areas, such as geometry, particularly with the assistance of symbolic methods (Trinh et al., 2024), the performance of LLMs remains limited on Olympiad-level problems (He et al., 2024). One of the key challenges in advancing competition-level math reasoning, compared to other domains like coding or grade-school math, is the scarcity of large-scale data. Creating valid and challenging math questions, along with providing correct solutions, is costly. This is especially true for Olympiad-level problems, which can be time-consuming even for experts. This highlights the need for scalable and automated methods to collect high-quality data for Olympiad-level problems to facilitate further advancements in this field.

On the evaluation side, in contrast to the rapid advancements in LLMs, the evaluation of their math reasoning capabilities remains relatively underdeveloped. First, as aforementioned, the cost

of creating and annotating advanced math problems is high. Second, popular datasets such as MATH (Hendrycks et al., 2021b) and GSM8K (Cobbe et al., 2021) have been saturated by both open-source and closed-source models (Achiam et al., 2023; Yang et al., 2024b). Third, benchmarks whose test sets are publicly available online (Hendrycks et al., 2021b; Cobbe et al., 2021; He et al., 2024; Zhang et al., 2023b) are prone to potential contamination. Although techniques like n-gram matching and locality-sensitive hashing have been applied as a common practice (Achiam et al., 2023; Dubey et al., 2024; Yang et al., 2024a) to reduce contamination, they still suffer low accuracy and would not be able to rule out rephrased questions, as shown by Yang et al. (2023). Given these limitations, it is crucial to develop an evolving evaluation benchmark that contains abundant and up-to-date test samples, and designed with appropriate difficulty to fairly assess a model's math reasoning abilities.

The Art of Problem Solving[1] (AoPS) forum is a rich resource for Olympiad-level math problems, featuring discussions on topics such as algebra, geometry, combinatorics, and number theory from competitions like AMC (AOPS, 2023), AIME (AOPS, 2024), and the International Mathematical Olympiad (IMO). However, the forum's unstructured nature, including irrelevant comments and incomplete solutions, poses challenges in extracting high-quality, structured question-answer (QA) pairs. Developing an effective automated pipeline to curate these QA pairs is essential to address the scarcity of large-scale, high-quality data for training and evaluating models in Olympiad-level math reasoning. In this paper, we utilize the posts from the AoPS forum to create a large-scale training and a contamination-resistant evaluation set. Our pipeline is designed to run automatically, enabling us to build and maintain evolving train/evaluation datasets. This automated approach is crucial, as it allows for continuously updating the datasets, ensuring they are less likely to suffer from contamination, even as existing data potentially becomes compromised over time. In summary, our key contributions are as follows:

- We build a pipeline to extract questions and solutions from raw AoPS forum data, constructing the AoPS-Instruct, a novel large-scale dataset with 666.1K Olympiad-level math QA pairs.

- Using the most recent QA pairs, We build an automatic pipeline that introduces LiveAoPS-Bench, a contamination-resistant evaluation set for assessing the math reasoning capabilities of LLMs.

- Our experiments on LiveAoPSBench show a declining performance trend over time for various LLMs, indicating potential data contamination, and stressing the need for up-to-date evaluation data.

- Fine-tuning various LLMs on AoPS-Instruct lead to improved performance on standard benchmarks such as OlympiadBench, Omni-Math, and our LiveAoPSBench dataset, verifying the effectiveness of our dataset in enhancing math reasoning capabilities of LLMs.

## 2 RELATED WORK

In this section, we provide an overview of the existing mathematical datasets used for evaluation and training purposes. Additionally, we review the latest methods and LLMs for enhancing and evaluating these math datasets.

**Evaluation Datasets for Math.** The evaluation of the mathematical capabilities of LLMs has traditionally relied on well-established and widely-used datasets such as GSM8K and MATH (Cobbe et al., 2021; Hendrycks et al., 2021b), which have served as benchmarks for several years. These datasets typically contain math problems ranging from middle-school to high-school level, providing broad coverage across various problem categories. However, they present two significant limitations: 1) being older, their test sets are more susceptible to contamination from current training data of LLMs (Yang et al., 2023), and 2) they have reached a level of saturation, with state-of-the-art (SOTA) models achieving over 90% accuracy (Yang et al., 2024b). To address these shortcomings, Zhang et al. (2023b) introduced the Gaokao dataset, which includes more challenging high school-level problems from the Chinese college entrance exam. In addition, newer datasets such as OlympiadBench (He et al., 2024), AMC23 (AOPS, 2023), AIME24 (AOPS, 2024), and Omni-Math

---

[1]https://artofproblemsolving.com/community

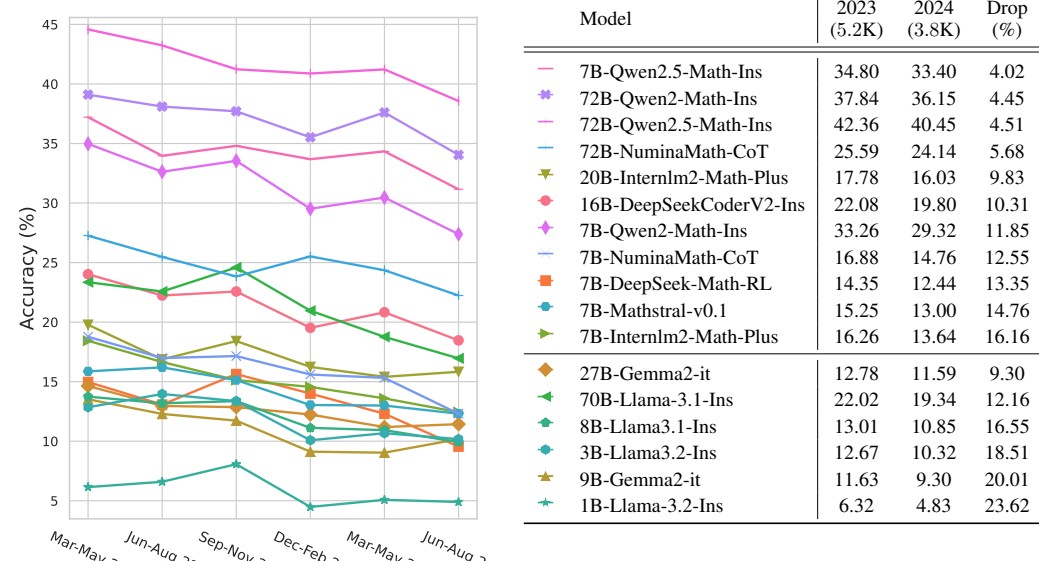

| Model | 2023 (5.2K) | 2024 (3.8K) | Drop (%) |
|---|---|---|---|
| 7B-Qwen2.5-Math-Ins | 34.80 | 33.40 | 4.02 |
| 72B-Qwen2-Math-Ins | 37.84 | 36.15 | 4.45 |
| 72B-Qwen2.5-Math-Ins | 42.36 | 40.45 | 4.51 |
| 72B-NuminaMath-CoT | 25.59 | 24.14 | 5.68 |
| 20B-Internlm2-Math-Plus | 17.78 | 16.03 | 9.83 |
| 16B-DeepSeekCoderV2-Ins | 22.08 | 19.80 | 10.31 |
| 7B-Qwen2-Math-Ins | 33.26 | 29.32 | 11.85 |
| 7B-NuminaMath-CoT | 16.88 | 14.76 | 12.55 |
| 7B-DeepSeek-Math-RL | 14.35 | 12.44 | 13.35 |
| 7B-Mathstral-v0.1 | 15.25 | 13.00 | 14.76 |
| 7B-Internlm2-Math-Plus | 16.26 | 13.64 | 16.16 |
| 27B-Gemma2-it | 12.78 | 11.59 | 9.30 |
| 70B-Llama-3.1-Ins | 22.02 | 19.34 | 12.16 |
| 8B-Llama3.1-Ins | 13.01 | 10.85 | 16.55 |
| 3B-Llama3.2-Ins | 12.67 | 10.32 | 18.51 |
| 9B-Gemma2-it | 11.63 | 9.30 | 20.01 |
| 1B-Llama-3.2-Ins | 6.32 | 4.83 | 23.62 |

Figure 1: Accuracy trends of various LLMs on LiveAoPSBench over an 18-month period, highlighting a consistent decline in performance. We saperate math expert model with general purpose model on the right. The degradation in accuracy varies across models, ranging from $2.4\%$ to $23.6\%$.

(Gao et al., 2024) represent higher levels of difficulty, collecting from more recent high school competition problems. While these datasets temporarily mitigate the risk of data contamination, they remain susceptible to this issue as LLMs continue to evolve, particularly with fine-tuning on newer data. To address this, we introduce LiveAoPSBench, which utilizes the most recent posts from the AoPS forum and applies substring-matching techniques to exclude any previously used problems from the new posts. More importantly, our pipeline is *fully automated*, allowing the evaluation set to evolve with forum posts, thereby significantly decreasing the likelihood of contamination.

**Training Datasets for Math.** Training datasets for mathematical reasoning can be categorized into two types: pretraining and supervised fine-tuning (SFT) datasets. First, pretraining datasets consist of large-scale math data, *e.g.*, billions of tokens used during the pretraining phase of LLMs. Notable examples include Open-Web-Math (Paster et al., 2024) and Minerva (Lewkowycz et al., 2022), which contain 38.5B and 14.7B tokens of math data, respectively. Second, SFT datasets focus on high-quality question-answer pairs. Examples include Open-Math-Instruct (Toshniwal et al., 2024), Orca-math (Mitra et al., 2024), and the training sets of widely used benchmarks such as GSM8K (Cobbe et al., 2021) and MATH (Hendrycks et al., 2021b). However, these datasets are generally limited to grade-school level mathematics and do not target more advanced topics like Olympiad-level math. One of the most closely related datasets to ours is Numina (Li et al., 2024), which combines popular SFT datasets like Orca-math, MATH, and GSM8K, along with approximately 190K new Olympiad-level QA pairs. Concurrently, Yue et al. (2024) introduced a large-scale instruction fine-tuning dataset for math and science, which has also shown improvements in mathematical reasoning. Table 1 presents a detailed comparison of our dataset with these related datasets.

**Contamination-Resistant Evaluation.** Benchmarks that are publicly accessible are prone to be contaminated due to the potential inadvertent data overlap during training. The typical decontamination method involves using exact substring (*e.g.*, $n$-gram) matching to detect overlaps with the target evaluation sets (Zhuo et al., 2024). However, this approach fails to catch rephrased examples and can not eliminate all overlaps with the test set.(Yang et al., 2023). While alternative LLM-based methods for decontamination have been proposed, they often lack guarantees and may result in high false-positive rates (Yang et al., 2023). A reliable way to mitigate contamination is to select data that appeared after LLMs were trained, known as the *knowledge cut-off*. In the code generation domain, LiveCodeBench (Jain et al., 2024) addresses this issue by categorizing data based on timestamps, setting a cutoff date, and designating data beyond this point as unseen. We adopt a similar strategy in the math domain, partitioning the dataset by timestamps and enabling users to select data based

Table 1: Comparison of our dataset with other related datasets from the literature. Our dataset uniquely includes timestamp information and leverages open-source large language models (LLMs)s like Qwen 2.5 72B for solution rewrites. $\star$ denotes inclusion of additional training datasets such as GSM8K, Orca-Math, and MATH. Datasets marked with $\dagger$ have their solutions entirely generated by LLM.

| Dataset | Dataset Size | | Time | Olympiad | Solution |
| | Train | Eval | Stamp | Level | Rewrite |
| --- | --- | --- | --- | --- | --- |
| Numina (Li et al., 2024) | 859K$\star$ | 0.1K | ✗ | ✓ | GPT4-o |
| OpenMathInstruct (Toshniwal et al., 2024) | 1.8M | – | ✗ | ✗ | Mixtral$^\dagger$ |
| OlympiadBench He et al. (2024) | – | 6.1K | ✗ | ✓ | Human |
| GSM8K (Cobbe et al., 2021) | 7.5K | 1K | ✗ | ✗ | Human |
| MATH (Hendrycks et al., 2021b) | 7.5K | 5K | ✗ | ✗ | Human |
| Orca-Math (Mitra et al., 2024) | 200K | - | ✗ | ✗ | GPT-4$^\dagger$ |
| AoPS (Ours) | 666.1K | 3.9K | ✓ | ✓ | Qwen 2.5 |

on specific dates. Although this approach may not fully eliminate rephrased existing questions, it ensures that evaluation data remains unseen and less contaminated, providing a more accurate and fair assessment of LLMs.

**Math-Specific Models.** Several specialized models have been developed to improve the mathematical reasoning capabilities of LLMs (Shao et al., 2024; Mistral, 2024; Li et al., 2024; Yang et al., 2024b; Azerbayev et al., 2024). These models are typically initialized from pretrained general-purpose models, trained on large math datasets, followed by math-specific SFT, and then refined through reinforcement learning with human feedback (RLHF). In this paper, we fine-tune both general and math-specific models to demonstrate that AoPS-Instruct brings consistent improvements.

## 3 AoPS Dataset

In this section, we first describe the process of extracting and cleaning QA pairs from the AoPS forum to construct our training set. Then we explain how to utilize the latest forum data to create a reliable, contamination-resistant evaluation dataset for assessing model performance.

### 3.1 Math Instruction Fine-tuning Dataset: AoPS-Instruct

We now describe the five steps of our automated pipeline for constructing the instruction fine-tuning dataset AoPS-Instruct.

**Step 0: Raw Forum Discussion Collection.** We begin by collecting raw discussions from the forum website, where each discussion is called a "topic". In these topics, the author presents math problems (*e.g.*, competition-level problems) or general questions, such as seeking advice or resources. Our raw dataset consists of $1,076,712$ topics. Topics posted up until December 2023 are used as the training set, while those posted between January and June 2024 are reserved as the evaluation dataset.

**Step 1: Math Question Detection.** We then filter out irrelevant topics, specifically those not containing a mathematical question. To achieve this, we use Llama-3.1 8B (Dubey et al., 2024) to decide the relevance of each topic. The first post of each topic determines whether the topic is a mathematical question or not, so we manually design a few-shot prompt, provide the first post of the topic to the model, and prompt the model to output if the topic is a math question or not. This step reduces the dataset to $478,337$ topics with math questions after pruning $598,375$ irrelevant ones.

**Step 2: Question-Answer Extraction.** After filtering, we extract the math question from the first post of each topic and identify potential solutions provided in subsequent posts. Since this task requires understanding the entire conversation and determining which responses contain valid solutions, we employ the 70B variant of Llama 3.1 for this step, enabling the detection of both the question and all relevant answers from the discussion.

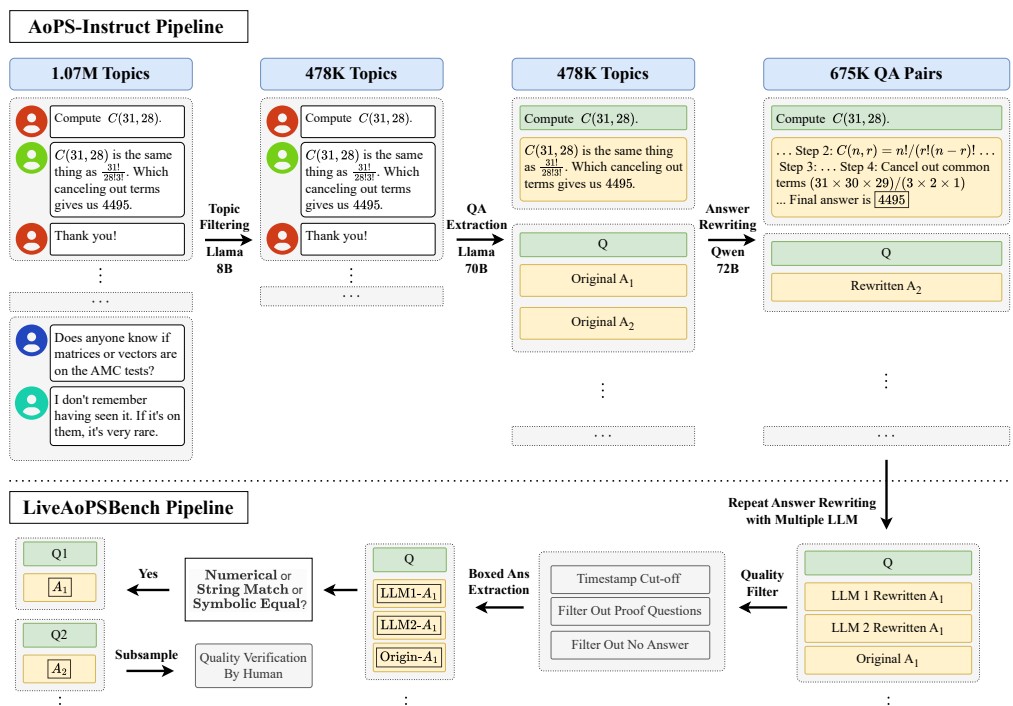

Figure 2: **The overall process of our dataset curation.** *Top:* Dataset cleaning pipeline (Training). First, irrelevant topics are detected by a small LLM, then we extract questions and answers from relevant discussions, and then each answer is rewritten into a step-by-step solution. *Bottom:* LiveAoPSBench curation pipeline (Evaluation). We take the most recent posts, use two LLMs to rewrite the solution, filter out the questions without clear final answer, and create the final evaluation set.

**Step 3: Solution rewriting.** Math solutions generated by users on the AOPS forum are often concise, omitting details assumed to be common knowledge among the target audience. For instance, a user might write $(x + yz)/2 \geq \sqrt{xyz}$ without explicitly mentioning the application of the AM-GM inequality to $(x, yz)$. While such brevity is typical for expert-level discussions, LLMs trained on these succinct solutions often struggle to maintain their chain-of-thought reasoning capabilities.

Our experiments show that fine-tuning a model on these concise solutions significantly degrades its performance on standard benchmarks (see Section 4.4 and Figure 5b). To address this issue, we utilize the Qwen 2.5 72B model (Yang et al., 2024b) to rewrite all solutions into detailed, step-by-step explanations. This approach aligns with similar techniques used in prior work, such as the Numina project (Li et al., 2024), which also employed solution rewriting to improve response quality. An example of a rewritten solution is provided in Figure 3, and the overall dataset curation process is illustrated in Figure 2.

**Step 4: Data Decontamination.** After processing all the QA pairs, we apply data decontamination to remove any overlap with the test sets of commonly used math benchmarks. Following the approach used in DeepSeekMath (Shao et al., 2024), we employ a 10-gram exact match decontamination (Zhuo et al., 2024) method to ensure that our dataset remains distinct from those benchmarks.

After following the steps described above, we have a total of 675K QA pairs, out of which $666,160$ are before Jan 2024 and constitute the AoPS-Instruct. We provide further statistics of our dataset in Section 4.1 and Figure 4.

## 3.2 CONTAMINATION-RESISTANT EVALUATION: LIVEAOPSBENCH

Math LLMs are trained on large instructional corpora. A common issue with current evaluation sets is the risk of contamination, where test samples may inadvertently overlap with training data. To

**Question:** Find all solutions in $\mathbb{N}$: $2^{2^x} + 1 = 17y$

**User Response:**
The multiplicative order of 2 modulo 17 is 8, since $2^4 = 16 \equiv -1 \pmod{17}$. Therefore, if $x \geq 3$, then $8 \mid 2^x$, and so $2^{2^x} \equiv 1 \pmod{17}$. So $x \leq 2$, and so the only solution is $(x, y) = (2, 1)$.

- - - - - - - - - - - - - - - - - - - - - - - - - - - - - - - - - - - - - - - - - - - - - - - -

**Qwen Rewritten Solution:**
1. Determine the multiplicative order of 2 modulo 17. We find that $2^4 = 16 \equiv -1 \pmod{17}$.
2. Since $2^4 \equiv -1 \pmod{17}$, it follows that $2^8 \equiv 1 \pmod{17}$. Therefore, the multiplicative order of 2 modulo 17 is 8.
3. Consider the equation $2^{2^x} + 1 = 17y$. If $x \geq 3$, then $2^x$ is a multiple of 8 (since $2^3 = 8$ and higher powers of 2 are multiples of 8).
4. If $2^x$ is a multiple of 8, then $2^{2^x} \equiv 1 \pmod{17}$, because the order of 2 modulo 17 is 8.
5. Substituting $2^{2^x} \equiv 1 \pmod{17}$ into the equation, we get $1+1 \equiv 17y \pmod{17}$, which simplifies to $2 \equiv 0 \pmod{17}$. This is a contradiction.
6. Therefore, $x$ must be less than 3. The possible values for $x$ are 0, 1, and 2.
7. Check each possible value of $x$:

   - For $x = 0$: $2^{2^0} + 1 = 2^1 + 1 = 3$, which is not divisible by 17.

   - For $x = 1$: $2^{2^1} + 1 = 2^2 + 1 = 5$, which is not divisible by 17.

   - For $x = 2$: $2^{2^2} + 1 = 2^4 + 1 = 16 + 1 = 17$, which is divisible by 17.

8. The only solution is $(x, y) = (2, 1)$.

Thus, the solution is $\boxed{(2, 1)}$.

Figure 3: Example of a rewritten solution for a posted problem and a user solution. The model fills in the trivial intermediate steps, reformats the response, and puts the final answer in a box.

create contamination-resistant benchmarks, we constructed our evaluation set by sorting the raw data based on the initial posting timestamp and including only the most recent entries. Our evaluation set, denoted as LiveAoPSBench, is sourced from the AoPS forum, with posts strictly between January 2023 and September 2024. We utilize the same pre-processing pipeline, depicted in Figure 2, to extract QA pairs and have the raw solutions rewritten for consistency.

**Filtering**. The correctness of the solution is typically verified by comparing the final answer to the human-annotated answer. Note that human-annotated answers may still contain errors, as we do not perform formal proofs or verification. When constructing an evaluation set, it is essential that each question has a concrete and definite answer, which is enclosed as $\boxed{ans}$ format for ease of parsing, as illustrated in Figure 3. We start by applying a series of heuristic filters to exclude proof-based questions and extract only those with explicit, boxed answers. To ensure that our test set does not contain problems included in widely used training sets, we use an stricter 8-gram matching filter—stricter compared to the 10-gram filter used for training set decontamination. This helps eliminate any potential overlap with common training corpora (Hendrycks et al., 2021b; Cobbe et al., 2021; Mitra et al., 2024).

**Cross-Check by LLMs**. A key challenge in building a fair evaluation set is ensuring the accuracy and validity of QA pairs. To automate this process, we employed two different models—Llama3.1-70B-Ins (Dubey et al., 2024) and Qwen2-72B-Ins (Yang et al., 2024a) to perform the rewriting step twice for each question. Consequently, for each question $Q$, we obtain a triplet: $(A_{\text{qwen}}, A_{\text{llama}}, A_{\text{original}})$. If a boxed answer is detected in $A_{\text{original}}$, it is automatically accepted as a candidate answer for the question. Following this, we performed a cross-check between $A_{\text{qwen}}$ and $A_{\text{llama}}$, removing all cases with inconsistent answers. This was done through string matching for text and value matching for numbers, while a SymPy-based (Meurer et al., 2017) symbolic equivalence program was used for SymPy-parsable expressions. The final answers are obtained by deduplicating the candidate answers. Through this process, we constructed LiveAoPSBench, which contains 3,863 examples. Further details can be found in Appendix A.

**Quality Verification.** We assess the quality of our dataset by having a group of 10 graduate students annotate a randomly selected 10% subset (386 cases) from our evaluation set. Each human annotator verifies whether the final answer is correct based on the raw post, with each question annotated by two different individuals. We report the percentage of cases marked as correct by the human annotators to measure the correlation between human judgment and our method. Additionally, since Olympiad-level questions can be challenging even for humans, we also report the inter-annotator agreement to evaluate consistency between different groups of human annotators. More details can be found in Section 4.4.

**Evolving Evaluation with Up-to-date Data.** Since our pipeline does not require human annotators, we are able to continuously update our LiveAoPSBench in an automated manner. This makes our benchmark an up-to-date and timestamped evaluation set that is resistant to contamination, thereby providing a more reliable mathematical evaluation resource for the research community.

## 4 EXPERIMENTS

### 4.1 DATASET STATISTICS

We provide a better overview of the AoPS dataset in Figure 4. As shown in Figure 4a, more than 60% of the questions have only one answer, while around 24% and 8% have two and three answers, respectively. Figure 4b shows the number of posts across each year, with a cut-off of August 2024. We observe that each year at least 15K mathematical questions are posted to the forum. This translates to more than 1,000 monthly questions, which shows the potential of the AoPS forum to be used as training, and especially evaluation set. Figure 4c shows a breakdown of the types of questions in our dataset. Proof questions and numerical questions with about 32% and 28% constitute the majority of the questions in our dataset.

Finally, Figure 4d shows the pairwise overlap between each pair of popular supervised fine-tuning datasets using substring matching between the two datasets of each pair. Among the two Olympiad-level datasets (*i.e.*, ours and Numina), our dataset has the least overlap with common datasets (with less than 14.1% overlap), which shows the number of new data points.

### 4.2 EVALUATING OPEN-SOURCED MODELS

We evaluate the models' performance as a function of time window. As shown in Fig 1, we find that all the models experience a performance drop when evaluating 2024 questions compared to questions in 2023. This decline suggests that performance on earlier examples may not accurately reflect the true capabilities of LLMs, as the initial results could be inflated by inadvertent data overlap.

### 4.3 INSTRUCTION FINE-TUNING

We show that the collected training dataset is effective at improving the math reasoning capabilities of LLMs. To this end, we choose 4 representative LLMs and fine-tune them on our dataset combined with the Numina (Li et al., 2024) dataset, and show that such a combination provides superior performance compared to training on either of the datasets alone.

We use the following set of diverse models for fine-tuning evaluation: (1) Mathstral-7B (Mistral, 2024): a math-specialized model derived from Mistral-7B (Jiang et al., 2023), (2) DeepSeekMath-7B (Shao et al., 2024): a math-specialized model based on the DeepSeek family, and (3) Llama 3.2 3B (Dubey et al., 2024) and (4) Llama 3.2 1B (Dubey et al., 2024), two recent general state-of-the-art models. For each QA pair, only the question is used as the instruction, with the rewritten solution serving as the response, formatted within the model's respective chat template. For instance, with Mathstral, we use the prompt: `[INST] question [/INST]solution` for instruction tuning.

Consistent with prior work, we train each model for three epochs (Shao et al., 2024; Yang et al., 2024b), as we observe additional epochs provide no further benefit (see Figure 9 in the Appendix for ablation studies on the number of epochs). We explore three data mixtures for fine-tuning: (1) AoPS alone, (2) Numina alone, and (3) AoPS + Numina. After fine-tuning each model, we eval-

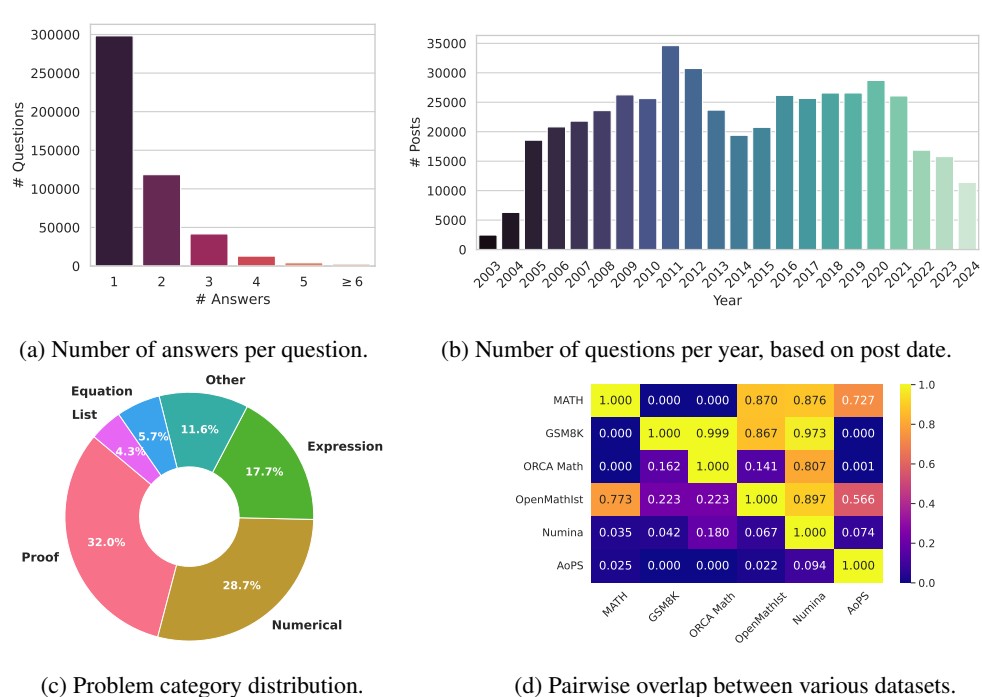

(a) Number of answers per question.

(b) Number of questions per year, based on post date.

(c) Problem category distribution.

(d) Pairwise overlap between various datasets.

Figure 4: AoPS Dataset Statistics. The statistics are across all the datapoints in our dataset before split. In (d), the percentage at row $i$ and column $j$ shows the fraction of the training set of $i$-th dataset (based on exact substring match) present in the $j$-th dataset. Our dataset has the least overlap with others with less than $14.1\%$ overlap.

uate the performance of each model on the following standard competition-level benchmarks: (1) OlympiadBench (He et al., 2024), which is an Olympiad-level evaluation dataset. Following prior literature (Yang et al., 2024a), we take only the math questions which have final answers and do not contain images or figures. This leaves us with 675 samples from this dataset (2) Omni-MATH (Gao et al., 2024), which is a collection of 4428 problems from various mathematical olympiad competitions. (3) LiveAoPSBench set for the year 2024. The results are shown in Table 2. As shown by the table, fine-tuning with our dataset consistently boost the performance.

## 4.4 ABLATION STUDIES

**Evaluation Quality Assessment**. We assess the quality of our evaluation set in two ways: by measuring its correlation with a well-established dataset and through manual evaluation over a subset of the data. First, He et al. (2024) compiled an Olympiad-level math evaluation set using manual assessment, which we leverage in our context to verify the quality of our method through the correlation between accuracies. Figure 5a, demonstrates that the evaluation on LiveAoPSBench is highly correlated with carefully established benchmarks such as OlympiadBench. This demonstrates that our automatically generated benchmark aligns closely with the quality of those created through extensive human effort. Next, we subsample 10% of our evaluation set and ask human annotators to verify the correctness of the final parsed answers by referring to the original post. Annotators are given three options: yes, no, and no-answer. "Yes" and "no" indicate whether the answer is deemed correct, while "no-answer" is selected when a concrete answer is not appropriate (*e.g.*, abstract concept questions answered with concrete examples). As a result, we found that 88% of the annotations were marked as correct, while 8% were incorrect and 4% fell under the no-answer category. To understand the gap from perfect accuracy here, we further measure the correlation between groups of human annotators by computing the percentage of choices that were consistent. Surprisingly, the human annotators only reached an agreement rate of 91%, demonstrating the challenge of evaluating Olympiad-level problems, even for graduate-level annotators.

Table 2: Performance comparison of different models fine-tuned on various datasets across multiple benchmarks. Bold values in the columns for No SFT, Numina, and AoPS-Ins represent the highest scores for individual datasets. Additionally, bold values for Numina+AoPS-Ins indicate performance that matches or surpasses all other fine-tuning alternatives. Our dataset outperforms Numina on most benchmarks, and the combined (Numina+AoPS-Ins) fine-tuning consistently yields superior results.

| Model | SFT Dataset | AoPS24 | Math | Olympiad Bench | Omni Math | AIME24 | AMC23 |
|---|---|---|---|---|---|---|---|
| Deepseek-Math 7b-Ins | No SFT | 11.7 | 47.1 | 14.5 | 12.3 | **1/30** | 8/40 |
| | Numina | 16.3 | 55.5 | **22.7** | 17.0 | 0/30 | 12/40 |
| | AoPS-Ins | **20.1** | **62.3** | 22.4 | **18.3** | 0/30 | **16/40** |
| | Numina+AoPS-Ins | **19.7** | **58.8** | **25.6** | **18.0** | **2/30** | 11/40 |
| Mathstral 7B | No SFT | 13.70 | 56.30 | 21.20 | 15.90 | 0/30 | **16/40** |
| | Numina | 15.70 | 54.60 | **23.40** | 17.10 | 0/30 | 15/40 |
| | AoPS-Ins | **22.40** | **60.30** | **23.40** | **17.60** | **1/30** | 14/40 |
| | Numina+AoPS-Ins | **23.50** | **60.60** | **27.30** | **20.10** | **2/30** | 14/40 |
| Llama-3.2 3B-Ins | No SFT | 12.0 | 47.4 | 16.1 | 12.9 | **2/30** | **11/40** |
| | Numina | 12.9 | 49.5 | **19.3** | 14.4 | 1/30 | 6/40 |
| | AoPS-Ins | **17.1** | **52.9** | 18.5 | **15.1** | 0/30 | **11/40** |
| | Numina+AoPS-Ins | **17.4** | **55.6** | **22.8** | **17.2** | 0/30 | 12/40 |
| Llama-3.2 1B-Ins | No SFT | 5.60 | 28.80 | 4.70 | 7.00 | 0/30 | 5/40 |
| | Numina | 6.90 | 32.70 | 6.40 | 9.70 | 0/30 | **6/40** |
| | AoPS-Ins | **8.60** | **34.70** | **11.10** | **11.00** | 0/30 | **6/40** |
| | Numina+AoPS-Ins | **10.50** | **36.60** | 10.40 | 10.30 | 0/30 | **6/40** |

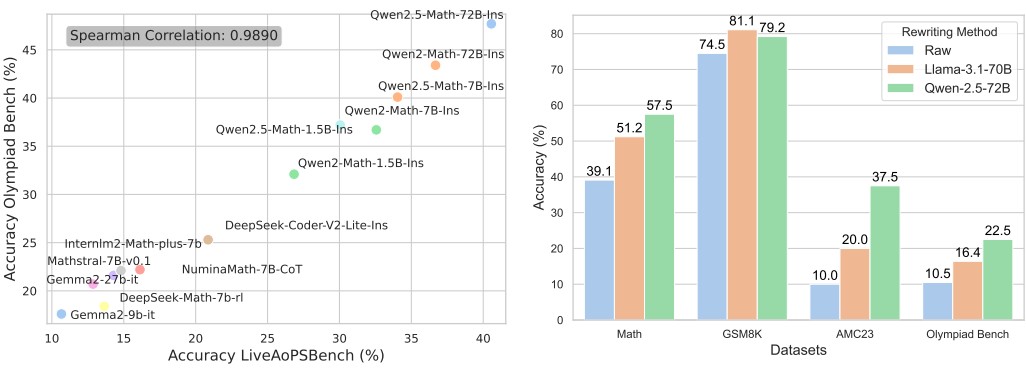

(a) Correlation with the OlympiadBench dataset.

(b) Ablation on Rewriting.

Figure 5: Ablations on LiveAoPSBench. (a) The performance of models on our benchmark is highly correlated with established datasets. (b) The effect of rewriting user solutions into a step-by-step solution with two different models. Rewriting solutions always improves accuracy, and using stronger models leads to larger accuracy gains.

**Rewritting's effect on performance.** We also ablate the effect of solution rewriting, which is an important part of our pipeline. As shown in Figure 5b, rewriting solutions into a step-by-step format substantially improves the test accuracy across all benchmarks. The Qwen-2.5 72B based rewriting performs favorably against Llama-3.1 70b based rewriting on competition-level math benchmarks, while being slightly worse on easier grade-school math. Overall, we found Qwen to be a stronger model, providing a higher amount of details and being less verbose compared to Llama in rewriting solutions (see Figure 17 in the Appendix for a qualitative example). This suggests that rewriting solutions with stronger models can significantly improve performance on benchmarks.

## 5 LIMITATIONS

**Absence of Visual Content.** Our dataset currently focuses on text-only problems, which may limit its effectiveness in certain areas, particularly geometry. Many geometry problems rely heavily on diagrams to fully convey the problem statement. Incorporating relevant images and figures could significantly enhance the dataset's comprehensiveness and applicability, especially in visually-dependent mathematical domains.

**Evaluation of Proof-based Questions.** Our evaluation dataset focuses on QA pairs with clear, final answers, which is well-suited to a broad range of Olympiad-level problems. However, a significant portion of such types of problems involve more complex proof-based questions that require detailed logical reasoning and multiple steps. While we incorporate proof-based questions in our instruction-tuning pipeline, the current evaluation pipeline lacks the ability to evaluate such questions effectively.

**Quality Variability in Community-Generated Content.** The community-driven content from the AoPS forum provides a rich source of high-quality data. Nevertheless, as with any community-generated content, the quality of answers and solutions can vary. While our filtering and refinement processes have successfully mitigated much of this noise, incorporating more advanced techniques in future iterations could result in better consistency and precision.

## 6 CONCLUSION AND FUTURE WORK

In conclusion, this paper introduces the AoPS-Instruct dataset and LiveAoPSBench, leveraging community-driven content from the Art of Problem-Solving forum to address the challenges of limited training data and unreliable evaluation for LLMs solving Olympiad-level math problems. By developing a scalable and automated pipeline for extracting and refining question-answer pairs, this work presents a dataset containing over $650,000$ QA pairs, along with an up-to-date, contamination-resistant evaluation benchmark. Our experiments demonstrate significant performance improvements across multiple standard benchmarks for models fine-tuned on the AoPS-Instruct, highlighting enhanced mathematical reasoning capabilities. Furthermore, the observed performance decline of various LLMs on LiveAoPSBench underscores the importance of continuously updating evaluation sets to mitigate the risks of data contamination.

For future work, there are several promising directions to explore. First, while this paper focuses on the AoPS forum, the pipeline developed is not limited to this domain. It is generalizable and can be applied to other online forums or different subject areas, enabling the creation of high-quality datasets for various fields, such as physics, computer science, or even non-technical disciplines. Expanding this pipeline to other knowledge-intensive communities could further improve the training and evaluation of LLM across disciplines. Additionally, the quality of the dataset can be significantly improved by incorporating more advanced LLMs into the pipeline. Leveraging state-of-the-art models for question extraction, answer detection, and solution rewriting would result in more accurate and detailed data, ultimately enhancing the effectiveness of the fine-tuned models. Lastly, the current pipeline focuses on question-answer pairs with clear final answers, but a significant portion of Olympiad-level problems involves proof-based questions that require a deeper evaluation of logical reasoning, argument structure, and intermediate steps. Future work could include adapting the pipeline to accommodate these proof-based problems, potentially using another advanced LLM as a judge (Li et al., 2023), or incorporating formalization methods to better assess these complex solutions.

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

# A MORE DETAILS ON LIVEAOPSBENCH

## A.1 EVALUATION PIPELINE STATISTICS

To begin with, we have 14158 QA pairs with time stamps between Jan-2024 and Aug-2024. Decontamination with 8-gram matching is performed against Math and GSM8K training set (Hendrycks et al., 2021b; Cobbe et al., 2021), which removes 664 Q-A pairs. After removing proof questions and non-boxed solutions, we are left with 7173 Q-A pairs over 5416 unique questions. Lastly, The LLM cross-check filters out 1553 questions with inconsistent solutions and the resulting LiveAoPS-Bench contains 3863 questions. We apply the same pipeline described in Sec 3.2 to data with a time stamp between Jan-2023 and Dec-2023 and get 5216 questions for the 2023 split result.

## A.2 HUMAN ANNOTATION

As shown in Figure 6, we develop a simple web interface for human annotators to verify the answers extracted by our LLMs. Annotators compare the "Voted Answer", "Original Answers" and all posts in the original topic page identified by LLMs to verify if the "Voted Answer" matches the original posts' answers. The verification process provides four results: Positive ("Yes"), negative ("No/No Answer"), and neutral ("Not sure"). The "Not sure" option is provided since verifying the answer sometimes requires a certain mathematical foundation and a significant amount of reading time. We also show highlight two examples of disagreement in Figure 7.

## A.3 DERIVATION OF DIFFICULTY LEVELS

The difficulty levels in this dataset do not reflect the exact difficulty of the problems but rather approximate the general education background of the problem, *e.g.*, this is a "High School" level problem. However, a challenging high school problem may be more complex than an easy college-level problem. The classification is derived from the problem tag in the AOPS forum, where the categories correspond to "Middle School", "High School", "College", and "High School Olympiads". In addition, some problems originate from special forums, which do not fit into the above categories and are classified as "Others" in our dataset.

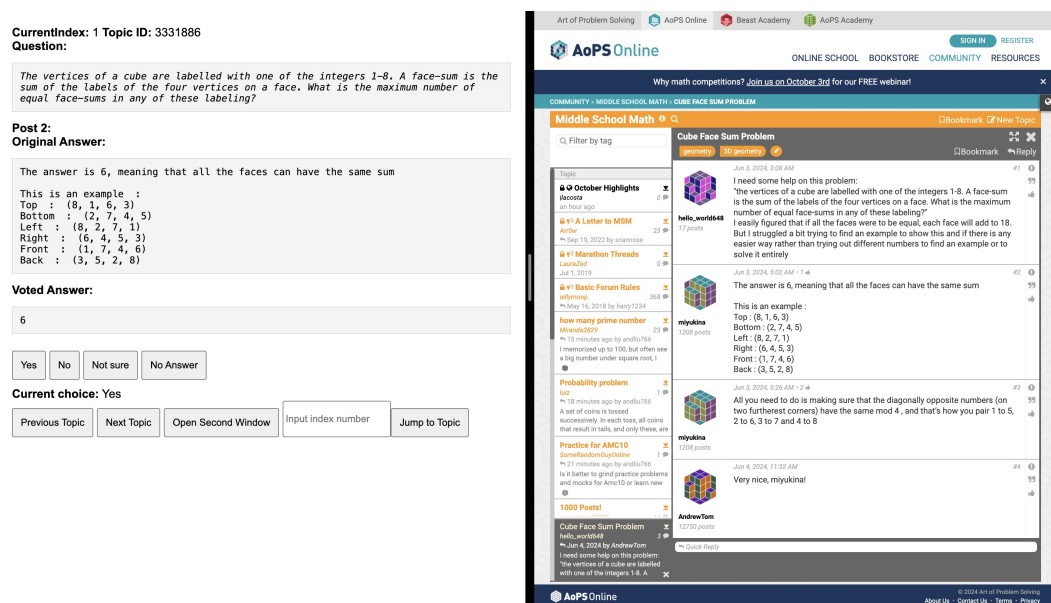

Figure 6: Human Annotation Interface.

**Example 1**

> **Question:** In a triangle, $ABC$, Angle $BAC = 90°$; $AD$ is the altitude from $A$ onto $BC$. Draw $DE$ perpendicular to $AC$ and $DF$ perpendicular to $AB$. Suppose $AB = 15$ and $BC = 25$. Then the length of $EF$ is?
>
> **Raw Post:**
> Because the triangle is a right triangle, so by the Pythagorean Theorem, the length of AC is 25² - 15² = 20. The area of ABC is AB * AC / 2 = 15 * 20 / 2. But it can also be represented by 25 * AD / 2. Putting them together we get 15 * 20 / 2 = 25 * AD / 2. So AD = 15 * 20 / 25 = 12. Because DE perpendicular to AC and DF perpendicular to AB, AEDF is a rectangle, which means that EF = AD, EF = 12. The answer is A.
>
> **Voted Answer**: 12
>
> - - - - - - - - - - - - - - - - - - - - - - - - - - - - - - - - - - - - - - - -
>
> **Is the Voted answer consistent with answer in raw post?**
> Human Annotator 1: ✓
> Human Annotator 2: ✗

**Example 2**

> **Question:** For a positive integer k we write $(1+x)(1+2x)(1+3x)......(1+kx) = a_0 + a_1x + a_2x^2 + ....... + a_kx^k$ where $a_0, ...a_k$ are the coefficients of the polynomial. Find the sum of all the digits of smallest possible value of k if $a_0 + a_1 + a_2 + ......a_(k-1)$ is divisible by 2005.
>
> **Raw Post:**
> $f(x) = (1+x)(1+2x)\ldots(1+kx) = a_0 + a_1x + \ldots + a_kx^k$
> $a_0 + a_1 + \ldots + a_{k-1} = f(1) - a_k$
> $a_k = 1 \cdot 2 \cdot 3 \ldots k = k!$
> $f(1) = 2 \cdot 3 \cdot 4 \ldots (1+k) = (k+1)!$
> $2005 \mid (k+1)! - k! \Longrightarrow 2005 \mid k \cdot k!$
> $2005 = 5 \cdot 401$
> $k \geq 401$
>
> **Voted Answer**: 5
>
> - - - - - - - - - - - - - - - - - - - - - - - - - - - - - - - - - - - - - - - -
>
> **Is the Voted answer consistent with answer in raw post?**
> Human Annotator 1: ✗
> Human Annotator 2: ✓

Figure 7: We highlight two examples of annotation inconsistencies caused by human annotators: 1. Example 1: Annotator 2 failed to recognize that the answer is explicitly stated in the raw post. 2. Example 2: The raw post does not directly provide the final answer. Annotator 1 was unable to reason that 4 + 0 + 1 = 5 constitutes the correct solution.

# B  DETAILED EVALUATION RESULTS ON LIVEAOPSBENCH

## B.1  EVALUATING OPEN-SOURCED LLMS

We have selected several mainstream open-source general LLMs and math-specific LLMs that demonstrate high performance on the previous math evaluation datasets. For math-specific LLMs, we choose DeepSeek-Math-7b-rl (Shao et al., 2024), Mathstral-7B-v0.1 (Mistral, 2024), 7b and 20b versions of Internlm2-Math-plus (Ying et al., 2024), 7B and 72B versions of NuminaMath-CoT (Li et al., 2024), 1.5B,7B,72B version of Qwen2-Math-Instruct (Yang et al., 2024a) and Qwen2.5-Math-Instruct (Yang et al., 2024b) as the representative of the math specific LLMs. Additionally, we include DeepSeek-Coder-V2-Lite-Instruct (Zhu et al., 2024), which is a code specialist model trained on both math and code corpus. For general purpose LLMs, We report performance on 1B, 3B and

| Model | 2023 | Jan | Feb | Mar | Apr | May | Jun | Jul | Aug | Sep | Oct | Nov | Dec |
|---|---|---|---|---|---|---|---|---|---|---|---|---|---|
| Count | 5216 | 483 | 388 | 444 | 415 | 472 | 412 | 396 | 505 | 381 | 409 | 404 | 507 |
| DeepSeek-Coder-V2-Lite-it | 22.45 | 23.40 | 19.33 | 22.97 | 22.17 | 22.67 | 21.12 | 23.99 | 25.74 | 22.83 | 23.72 | 22.28 | 18.93 |
| DeepSeek-Math-7b-rl | 15.38 | 18.22 | 15.46 | 17.57 | 13.49 | 14.41 | 12.86 | 15.66 | 14.65 | 15.75 | 15.89 | 17.08 | 13.61 |
| Internlm2-Math-plus-20b | 18.23 | 18.84 | 16.75 | 17.79 | 18.31 | 20.55 | 17.72 | 16.41 | 18.42 | 17.06 | 21.27 | 18.81 | 16.57 |
| Internlm2-Math-plus-7b | 17.10 | 17.81 | 16.49 | 19.82 | 17.11 | 18.01 | 15.05 | 19.95 | 16.24 | 13.91 | 18.58 | 17.08 | 15.19 |
| Mathstral-7B-v0.1 | 15.91 | 17.18 | 14.95 | 15.54 | 14.22 | 16.31 | 16.75 | 16.67 | 18.22 | 13.91 | 14.18 | 18.32 | 14.20 |
| NuminaMath-72B-CoT | 26.15 | 29.40 | 25.77 | 23.87 | 25.30 | 29.87 | 22.57 | 25.51 | 25.54 | 24.41 | 27.87 | 25.50 | 27.02 |
| NuminaMath-7B-CoT | 17.29 | 18.43 | 14.95 | 18.24 | 15.90 | 19.70 | 15.53 | 17.68 | 17.43 | 15.22 | 18.83 | 19.80 | 15.38 |
| Qwen2-Math-1.5B-it | 29.06 | 31.47 | 26.55 | 31.31 | 31.08 | 28.81 | 26.21 | 26.77 | 29.90 | 27.82 | 29.58 | 32.43 | 26.43 |
| Qwen2-Math-72B-it | 37.96 | 41.41 | 36.34 | 37.39 | 38.31 | 38.35 | 37.14 | 37.37 | 38.61 | 34.91 | 40.10 | 39.60 | 35.50 |
| Qwen2-Math-7B-it | 33.07 | 33.13 | 35.31 | 33.33 | 31.33 | 36.86 | 28.64 | 35.10 | 33.27 | 30.45 | 32.52 | 36.39 | 30.57 |
| Qwen2.5-Math-1.5B-it | 34.72 | 36.02 | 32.47 | 35.81 | 34.94 | 37.50 | 32.04 | 31.31 | 36.04 | 33.86 | 36.67 | 35.15 | 33.73 |
| Qwen2.5-Math-72B-it | 42.04 | 44.31 | 38.40 | 41.22 | 41.93 | 45.55 | 44.17 | 40.15 | 42.18 | 35.43 | 46.21 | 43.32 | 40.43 |
| Qwen2.5-Math-7B-it | 34.87 | 35.82 | 31.70 | 36.26 | 35.42 | 37.71 | 29.61 | 33.59 | 37.43 | 32.28 | 34.72 | 37.87 | 34.52 |
| Llama-3.2-1B-it | 6.75 | 6.83 | 5.93 | 4.73 | 5.54 | 9.53 | 5.10 | 8.08 | 6.73 | 8.66 | 7.09 | 7.92 | 5.13 |
| Llama-3.2-3B-it | 13.77 | 14.49 | 13.40 | 12.16 | 13.98 | 14.19 | 13.11 | 16.92 | 12.08 | 14.44 | 15.16 | 13.61 | 12.43 |
| Llama-3.1-8B-it | 14.03 | 15.53 | 13.66 | 14.86 | 13.73 | 13.56 | 11.65 | 15.15 | 16.04 | 12.86 | 14.91 | 14.60 | 11.64 |
| Gemma-2-27b-it | 13.80 | 11.39 | 13.92 | 15.77 | 13.01 | 15.04 | 10.68 | 14.90 | 15.64 | 12.86 | 14.91 | 14.36 | 13.02 |
| Gemma-2-9b-it | 12.42 | 10.97 | 10.05 | 13.51 | 12.53 | 13.35 | 11.17 | 13.38 | 15.05 | 10.76 | 13.45 | 13.61 | 10.85 |

| Model | 2024 | Jan | Feb | Mar | Apr | May | Jun | Jul | Aug |
|---|---|---|---|---|---|---|---|---|---|
| Count | 3863 | 634 | 527 | 614 | 503 | 511 | 380 | 363 | 331 |
| DeepSeek-Coder-V2-Lite-it | 20.86 | 18.14 | 22.77 | 25.41 | 22.47 | 18.00 | 17.89 | 21.21 | 19.64 |
| DeepSeek-Math-7b-rl | 13.64 | 14.04 | 17.08 | 15.31 | 15.71 | 10.37 | 10.53 | 13.50 | 9.97 |
| Internlm2-Math-plus-20b | 16.93 | 14.51 | 22.01 | 17.59 | 17.50 | 14.48 | 14.74 | 18.73 | 15.71 |
| Internlm2-Math-plus-7b | 14.81 | 11.83 | 18.60 | 17.59 | 16.50 | 11.15 | 17.11 | 13.77 | 10.88 |
| Mathstral-7B-v0.1 | 14.29 | 11.04 | 17.65 | 17.43 | 16.50 | 10.96 | 12.11 | 12.95 | 15.11 |
| NuminaMath-72B-CoT | 24.95 | 22.08 | 28.65 | 28.66 | 23.46 | 23.48 | 22.37 | 27.55 | 22.36 |
| NuminaMath-7B-CoT | 16.13 | 13.56 | 20.87 | 17.10 | 19.68 | 13.89 | 13.42 | 16.25 | 12.69 |
| Qwen2-Math-1.5B-it | 26.84 | 25.24 | 29.79 | 30.46 | 27.24 | 22.90 | 25.79 | 28.37 | 23.56 |
| Qwen2-Math-72B-it | 36.68 | 34.38 | 37.57 | 41.69 | 36.98 | 36.59 | 35.79 | 36.91 | 30.82 |
| Qwen2-Math-7B-it | 30.05 | 28.08 | 31.69 | 34.20 | 31.41 | 27.98 | 29.74 | 30.85 | 24.17 |
| Qwen2.5-Math-1.5B-it | 32.57 | 28.71 | 33.59 | 39.25 | 31.21 | 31.70 | 30.00 | 34.99 | 29.61 |
| Qwen2.5-Math-72B-it | 40.56 | 40.54 | 41.56 | 45.11 | 40.16 | 37.96 | 40.53 | 42.42 | 33.23 |
| Qwen2.5-Math-7B-it | 34.04 | 32.18 | 35.48 | 41.37 | 32.41 | 32.09 | 32.11 | 34.99 | 28.40 |
| Llama-3.2-1B-it | 5.80 | 3.63 | 7.40 | 6.84 | 6.16 | 4.70 | 5.53 | 7.16 | 5.44 |
| Llama-3.2-3B-it | 11.75 | 8.68 | 13.85 | 12.87 | 14.12 | 10.18 | 13.42 | 9.92 | 11.18 |
| Llama-3.1-8B-it | 12.71 | 10.57 | 16.13 | 14.17 | 11.33 | 12.33 | 12.63 | 12.67 | 11.48 |
| Gemma-2-27b-it | 12.87 | 10.57 | 16.13 | 13.03 | 13.52 | 11.94 | 11.58 | 14.05 | 12.39 |
| Gemma-2-9b-it | 10.67 | 8.68 | 11.95 | 13.36 | 10.93 | 8.02 | 8.16 | 14.05 | 10.27 |

Table 3: Accuracy per Month for Different Models

8B versions of the Llama3 family models (Dubey et al., 2024) as well as 9B and 27B versions of Gemma-2-Instruct (Team et al., 2024) model.

## B.2 DETAILED RESULTS

The accuracy comparison for these mainstream open source LLMs are shown in Tables 3, 4, 5 split by Month, Difficulty and Answer Type. The Month tables separately include evaluation results for 2023 and 2024. For the Difficulty and Answer Type tables, we use only the most recent evaluation results from 2024. Notably, the difficulty labels represent the general educational background of the problems rather than their exact difficulty. Over half of the problems originate from educational backgrounds associated with High School or High School Olympiads, and only around 7% are from Middle School, indicating our dataset's focus is more on the complex problems. Similarly, in the Answer Type Table, more than half of the problems are categorized as numeric-int.

## C TRAINING SET DETAILS

### C.1 DECONTAMINATION DETAILS

We use 10-gram substring matching to decontaminate against test set for a comprehensive list of math evaluation datasets available. (Cobbe et al., 2021; Hendrycks et al., 2021b; He et al., 2024; AOPS, 2023; 2024; Zhang et al., 2023b; Lewkowycz et al., 2022; Gao et al., 2024; Miao et al., 2020; Hendrycks et al., 2021a; Koncel-Kedziorski et al., 2016; Patel et al., 2021; Zhang et al., 2023a). In Figure 8. We show the decontamination statistic for our dataset and Numina.

Table 4: Accuracy per Difficulty for Different Models: The difficulty labels are for general education background of the problem and do not reflect the exact difficulty of the problem.

| Model | Overall | Middle School | High School | College | High School Olympiads | Others |
|---|---|---|---|---|---|---|
| Count | 3863 | 286 | 1349 | 314 | 889 | 1025 |
| DeepSeek-Coder-V2-Lite-it | 20.86 | 24.48 | 19.79 | 22.93 | 17.21 | 23.80 |
| DeepSeek-Math-7b-rl | 13.64 | 22.73 | 12.23 | 14.97 | 8.55 | 16.98 |
| Internlm2-Math-plus-20b | 16.93 | 24.83 | 15.64 | 19.11 | 12.71 | 19.41 |
| Internlm2-Math-plus-7b | 14.81 | 20.63 | 13.94 | 17.20 | 9.67 | 18.05 |
| Mathstral-7B-v0.1 | 14.29 | 19.23 | 12.90 | 15.61 | 10.24 | 17.85 |
| NuminaMath-72B-CoT | 24.95 | 33.57 | 25.28 | 26.75 | 18.45 | 27.22 |
| NuminaMath-7B-CoT | 16.13 | 18.18 | 15.12 | 16.24 | 12.60 | 19.90 |
| Qwen2-Math-1.5B-it | 26.84 | 32.17 | 26.17 | 28.98 | 22.61 | 29.27 |
| Qwen2-Math-72B-it | 36.68 | 43.36 | 38.62 | 42.04 | 28.01 | 38.15 |
| Qwen2-Math-7B-it | 30.05 | 37.06 | 30.17 | 31.21 | 24.86 | 32.10 |
| Qwen2.5-Math-1.5B-it | 32.57 | 38.81 | 33.65 | 31.85 | 28.91 | 32.78 |
| Qwen2.5-Math-72B-it | 40.56 | 48.25 | 42.70 | 45.86 | 32.62 | 40.88 |
| Qwen2.5-Math-7B-it | 34.04 | 42.66 | 34.84 | 36.62 | 27.11 | 35.80 |
| Llama-3.2-1B-it | 5.80 | 10.14 | 4.30 | 5.41 | 3.71 | 8.49 |
| Llama-3.2-3B-it | 11.75 | 18.53 | 10.23 | 9.55 | 8.10 | 15.71 |
| Llama-3.1-8B-it | 12.71 | 17.48 | 10.60 | 15.29 | 7.54 | 17.85 |
| Gemma-2-27b-it | 12.87 | 20.63 | 11.86 | 13.38 | 7.65 | 16.39 |
| Gemma-2-9b-it | 10.67 | 15.38 | 8.30 | 12.74 | 7.65 | 14.44 |

Table 5: Accuracy per Answer Type for Different Models: As not all answers can be easily verified, we divide the answers into different types to facilitate more accurate comparison and more convenient observation of the structural distribution of the dataset.

| Model | Overall | equation | expression | list | numeric-dec | numeric-int | numeric-irr | others |
|---|---|---|---|---|---|---|---|---|
| Count | 3863 | 296 | 950 | 195 | 57 | 2114 | 176 | 75 |
| DeepSeek-Coder-V2-Lite-it | 20.86 | 18.24 | 16.00 | 20.00 | 15.79 | 24.36 | 11.93 | 21.33 |
| DeepSeek-Math-7b-rl | 13.64 | 11.15 | 10.42 | 10.77 | 8.77 | 16.18 | 6.25 | 21.33 |
| Internlm2-Math-plus-20b | 16.93 | 14.53 | 12.74 | 13.33 | 14.04 | 20.20 | 8.52 | 18.67 |
| Internlm2-Math-plus-7b | 14.81 | 11.49 | 9.26 | 12.31 | 12.28 | 18.31 | 8.52 | 22.67 |
| Mathstral-7B-v0.1 | 14.29 | 14.53 | 10.74 | 6.67 | 12.28 | 16.65 | 10.80 | 21.33 |
| NuminaMath-72B-CoT | 24.95 | 19.59 | 20.21 | 16.92 | 24.56 | 28.71 | 20.45 | 32.00 |
| NuminaMath-7B-CoT | 16.13 | 14.86 | 12.11 | 11.28 | 12.28 | 18.78 | 12.50 | 21.33 |
| Qwen2-Math-1.5B-it | 26.84 | 23.31 | 22.11 | 28.72 | 15.79 | 29.52 | 25.57 | 32.00 |
| Qwen2-Math-72B-it | 36.68 | 27.70 | 30.84 | 36.41 | 26.32 | 41.15 | 31.82 | 40.00 |
| Qwen2-Math-7B-it | 30.05 | 23.99 | 26.21 | 27.69 | 19.30 | 33.30 | 25.00 | 37.33 |
| Qwen2.5-Math-1.5B-it | 32.57 | 25.34 | 28.74 | 32.82 | 22.81 | 35.86 | 22.73 | 46.67 |
| Qwen2.5-Math-72B-it | 40.56 | 31.42 | 38.32 | 41.54 | 31.58 | 43.19 | 36.36 | 45.33 |
| Qwen2.5-Math-7B-it | 34.04 | 28.72 | 30.21 | 33.85 | 24.56 | 36.90 | 30.11 | 40.00 |
| Llama-3.2-1B-it | 5.80 | 2.70 | 3.79 | 2.56 | 5.26 | 7.66 | 3.41 | 5.33 |
| Llama-3.2-3B-it | 11.75 | 7.77 | 8.32 | 7.18 | 7.02 | 14.71 | 6.25 | 16.00 |
| Llama-3.1-8B-it | 12.71 | 4.05 | 9.58 | 10.77 | 17.54 | 15.33 | 9.66 | 21.33 |
| Gemma-2-27b-it | 12.87 | 7.77 | 10.11 | 9.74 | 14.04 | 15.28 | 9.09 | 16.00 |
| Gemma-2-9b-it | 10.67 | 7.09 | 7.89 | 8.72 | 14.04 | 12.54 | 7.95 | 16.00 |

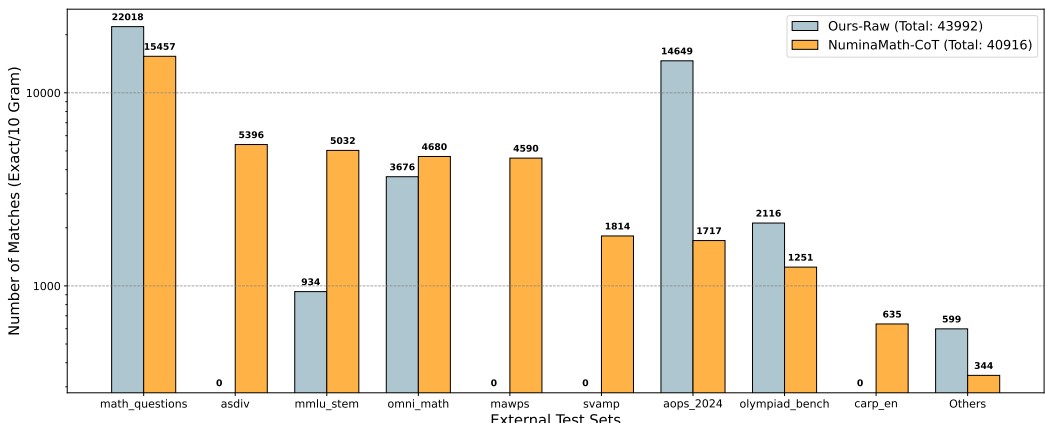

Figure 8: Decontamination Statistics: We perform decontamination on the raw dataset to produce AoPS-Instruct, with the same method as the Numina-Math-COT. Both datasets show considerable overlap with the MATH dataset. AoPS-Instruct exhibits more contamination within our 2024 split due to repeated questions, while Numina-Math-COT has higher contamination with other external datasets, reflecting its multi-source composition.

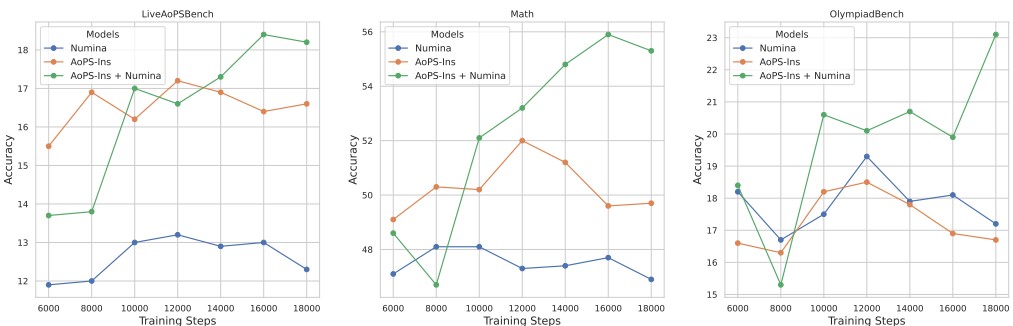

Figure 9: Ablation study on accuracy with respect to training steps. Here, 18,000 steps approximately correspond to 6 epochs for AoPS-Instruct and Numina, and 3 epochs for AoPS-Instruct + Numina. We can see that LiveAoPSBench + Numina consistenly improve as training goes on.

# D SFT EXPERIEMENTS

## D.1 ABLATION WITH CONTROLLED COMPUTATION BUDGET

As shown in Tab 2, Numina + AoPS-Instruct performs favorably against using AoPS-Instruct or Numina alone. To show this gain is not simply achieved by doubling the computation available for fine-tuning. We control the total fine-tune budget the same for AoPS-Instruct only, numina only and AoPS-Instruct + numina. This results in approximately 6 epoch on AoPS-Instruct or Numina or 3 epoch of training on AoPS-Instruct + Numina. We show the curve of ACC on Math,LiveAoPSBench, OlympiadBench w.r.t. training steps.

## D.2 REWRITING MODEL ABLATION

We use Qwen 2.5 72B to rewrite the solutions, and then we fine-tune smaller models on our dataset. This may raise the question of whether the effectiveness of our dataset would be limited by the capabilities of its rewriting model. To show the effectiveness of our dataset, we use a Qwen 2.5 1.5B to rewrite the solutions and then fine-tune DeepSeek-Math 7B-instruct on the dataset. Table

Table 6: Performance comparison of original DeepSeek-Math, Qwen2.5-1.5B, and DeepSeek-Math fine-tuned on solutions rewritten by Qwen2.5-1.5B. The fine-tuned DeepSeek-Math significantly outperforms both the original model and the rewriting model, demonstrating that our dataset enhances reasoning capabilities beyond the limitations of its rewriting model.

| Model | AIME24 | AMC23 | Olympiad Bench | Math | AoPS24 | Omni Math |
|---|---|---|---|---|---|---|
| Deekseek-Math-7b-Ins | **1/30** | 8/40 | 14.5 | 47.1 | 11.7 | 12.3 |
| Qwen2.5-1.5b-Ins | 0/30 | 9/40 | 21.3 | 55.0 | 16.7 | 16.8 |
| Deepseek-Math-7b-Ins (fine-tuned) | **1/30** | **13/40** | **22.7** | **61.0** | **19.4** | **19.2** |

---

**Insturction:**
You are given an online Math QA post. Your task is to identify whether the post asked is a concrete mathematical question, note that this means it shouldn't be an abstract general question related to math, and output the result as \boxed{0} for no and \boxed{1} for yes. A few examples are provided below:

*Few shot examples...*

Now, your task is to provide output for the following post:

Post: *Post 1*

- - - - - - - - - - - - - - - - - - - - - - - - - - - - - - - - - - - - - - - - - - - - - - - - - -

**Example Classify result:**

\boxed{*0/1*}

---

Figure 10: Prompt for the Topic Filtering part in Fig 2.

6 shows the performance of the original DeepSeek-Math, the performance of Qwen2.5-1.5B, and the performance of fine-tuned DeepSeek on Qwen2.5-1B-rewritten solutions. As shown by the results, the fine-tuned version outperforms both models, which shows that our dataset can improve the reasoning capability beyond its rewriting model solution.

# E    USE OF AoPS AS A DATA SOURCE

Concurrent to our work, both Numina (Li et al., 2024) and Omni-math (Gao et al., 2024) also use AoPS as their data source. Different from us, Numina only includes data from the contest page with 30K questions [2], while we utilize all the 1.07 available posts on this forum. Furthermore, Omni-math (Gao et al., 2024) includes only 4428 evaluation questions from all timestamps, while we include the *most recent* problems posted in 2024, as well as a *large-scale* training set.

# F    PROMPTS

We provide the Prompts used in our pipeline in Figures 10, 11, and 12.

# G    DATASET EXAMPLES

We provide further examples of our dataset and its rewritten solutions in Figures 13, 14, 15, 16. 17. and 18.

---

[2]https://artofproblemsolving.com/community/c13_contest_collections

**Insturction:**
You are given an online Math QA forum where each user post in each topic is in the format "post i by user j: [post i text]". Each user may reply to other users by quoting their post. Your task is to identify the question asked by the first user and find all potential answers within the follow-up discussion, and output them in a structured json format.

Your output json must have a "question" key containing the question, and one "answers" key containing the list of answers. Each answer must have three keys: a "user" key to identify the user who posted the solution, a "post number" to identify which post number the answer originates from, and a "content" key for the content of the solution. Make sure to reformat the answer content to make it a formal clean solution, without missing details. Do not include any irrelevant information in the answer. Do not add any additional information to the question or answers.

Ensure to handle different line breaks and spaces between posts accurately, and maintain the sequence of the dialogue. Always surround mathematical questions with $ symbols for LaTeX formatting. In case the dialogou does not contain any mathematical question, or there are no valid answers, leave the "question" or "answer" key empty. A few examples are proided below:

*Few shot examples...*

Now, your task is to provide JSON output for the following Topic:

*post 1 by user1: ...*
*post 2 by user2: ...*

---

**Example Parse result:**
```
{
    "question": "Question from Post 1",
    "answers": [
        {
            "user": "User2",
            "post number": 2,
            "content": "Solution 1"
        },
        {
            "user": "User4",
            "post number": 5,
            "content": "Solution 2"
        }"
    ]
}
```

Figure 11: Prompt for the QA extraction part in Fig 2.

**Insturction:**

You are given a solution to a mathematical question. Your task is to re-write the solution into a step-by-step solution with itemized steps(1..., 2...., 3....). You should re-write the solution in a formal and clean way, without missing any details. Make sure to include all the necessary steps and explanations. Do not include any irrelevant information in the answer. Do not add any additional information to the solution. Always surround mathematical questions with $ symbols for LaTeX formatting. If the question contains a final number, make sure to put the number in a

boxed for LaTeX formatting.

Question: **Question**

Solution: **Raw Answer**

Now provide the formalized re-written answer. DO NOT include the question in your answer. Only respond with the re-written solution.

- - - - - - - - - - - - - - - - - - - - - - - - - - -

**Example Rewrite result:**
**Rewritten solutions**

Figure 12: Prompt for the Solution Rewriting Part in Fig 2.

**Question:** Find all pairs $(k, l)$ of positive integers such that $\frac{k^l}{l^k} = \frac{k!}{l!}$

**User Response 1:**
W.l.o.g we may assume $k \geq l$.
If $k \geq l + 2$, then $k - 1$ divides the RHS, thus the LHS, thus $k^l$, but $k, k - 1$ are coprime, thus this is only possible if $k = 2$, which would imply $l = 0$, contradiction.
So the cases $k = l, l + 1$ are left.
If $k = l$, then we always have equality.
If $k = l + 1$, then the we need $\frac{k^{k-1}}{(k-1)^k} = k \iff k^{k-2} = (k-1)^k$.
Again $k - 1$ is supposed to divide a power of $k$, thus we remain with $k = 2$, so with $l = 1$, clearly being a solution.

**User Response 2:** If (k,l) solution, then (l,k) is solution too. Therefore we can find solutions $k \geq l$. If l=1 we have $k = k! \implies k = 1$ or $k = 2$.
If l=2 we have $k^2 = 2^{k-1}k!$ solution is k=2.
If $l \geq 3$ then $\frac{k^l}{l^k} \leq 1$ and $\frac{k!}{l!} \geq 1$, therefore solution is l=k.
All solutions $(k, l) = (1, 2)(2, 1), (n, n), n \in N$.

- - - - - - - - - - - - - - - - - - - - - - - - - - - - - - - - - - - -

**Qwen Rewritten Solution 1:**
1. Without loss of generality, assume $k \geq l$.
2. Consider the case $k \geq l + 2$:

- The right-hand side (RHS) of the equation $\frac{k^l}{l^k} = \frac{k!}{l!}$ is divisible by $k - 1$.
- Therefore, the left-hand side (LHS) must also be divisible by $k - 1$.
- Since $k$ and $k - 1$ are coprime, $k - 1$ must divide $k^l$.
- This is only possible if $k = 2$, which would imply $l = 0$, leading to a contradiction.

3. Therefore, the only possible cases are $k = l$ and $k = l + 1$.
4. If $k = l$:

- The equation $\frac{k^l}{l^k} = \frac{k!}{l!}$ simplifies to $\frac{k^k}{k^k} = \frac{k!}{k!}$, which is always true.

5. If $k = l + 1$:

- The equation becomes $\frac{(l+1)^l}{l^{l+1}} = \frac{(l+1)!}{l!}$.
- Simplifying, we get $\frac{(l+1)^l}{l^{l+1}} = l + 1$.
- This further simplifies to $(l+1)^{l-1} = l^{l+1}$.
- Again, $l$ and $l + 1$ are coprime, so $l$ must divide a power of $l + 1$.
- The only solution is $l = 1$, which implies $k = 2$.

6. Therefore, the pairs $(k, l)$ that satisfy the equation are $\boxed{(k, l) = (k, k)}$ and $\boxed{(k, l) = (2, 1)}$.

**Qwen Rewritten Solution 2:**
1. If $(k, l)$ is a solution, then $(l, k)$ is also a solution. Therefore, we can assume without loss of generality that $k \geq l$.
2. Consider the case when $l = 1$. The equation becomes $\frac{k^1}{1^k} = \frac{k!}{1!}$, which simplifies to $k = k!$. The only positive integer solutions to this equation are $k = 1$ and $k = 2$.
3. Consider the case when $l = 2$. The equation becomes $\frac{k^2}{2^k} = \frac{k!}{2!}$. This simplifies to $k^2 = 2^{k-1}k!$. The only positive integer solution to this equation is $k = 2$.
4. Consider the case when $l \geq 3$. We have $\frac{k^l}{l^k} \leq 1$ and $\frac{k!}{l!} \geq 1$. For these two inequalities to hold simultaneously, it must be that $k = l$.
5. Therefore, the solutions are $(k, l) = (1, 2), (2, 1)$, and $(n, n)$ for any positive integer $n$.

Figure 13: An example illustrating two distinct rewritten solutions for a single posted problem, where each user has proposed a unique approach to solving it.

**Question:** Let $ABC$ be an equilateral triangle. On the plane $(ABC)$ rise the perpendiculars $AA'$ and $BB'$ on the same side of the plane, so that $AA' = AB$ and $BB' = \frac{1}{2}AB$. Determine the measure the angle between the planes $(ABC)$ and $(A'B'C')$.

**User Response:**
Denote $A'B' \cap AB = D$. Then $BD = BA = BC$ so $\angle ACD = 90°$. Thus the plane $AA'C$ is perpendicular to $CD$ and the angle between $A'B'C$ and $ABC$ is equal to $\angle A'CA = 45°$.

- - - - - - - - - - - - - - - - - - - - - - - - - - - - - - - - - - - - - - - - - - - - - - - - - - - - - - - -

**Qwen Rewritten Solution:**
1. Denote the intersection of $A'B'$ and $AB$ as $D$.
2. Since $BD = BA = BC$, it follows that $\triangle BDA$ and $\triangle BDC$ are isosceles triangles.
3. Therefore, $\angle ACD = 90°$.
4. The plane $AA'C$ is perpendicular to $CD$.
5. The angle between the planes $(A'B'C')$ and $(ABC)$ is equal to $\angle A'CA$.
6. Since $AA' = AB$ and $BB' = \frac{1}{2}AB$, and given the symmetry and properties of the equilateral triangle, $\angle A'CA = 45°$.
7. Thus, the measure of the angle between the planes $(ABC)$ and $(A'B'C')$ is $\boxed{45°}$.

Figure 14: An example of a rewritten geometry solution for a posted problem.

**Question:** Solve the equation in parameter $p$: $x^3 + px^2 + (p - 1 + \frac{1}{p-1})x + 1 = 0$

**User Response:** Note that $p \neq 1$. Then $x = 1 - p$ works, you can get the other roots (when they exist) by quadratic formula.

- - - - - - - - - - - - - - - - - - - - - - - - - - - - - - - - - - - - - - - - - -

**Qwen Rewritten Solution:**

1. First, observe that the parameter $p$ must satisfy $p \neq 1$ to avoid division by zero in the term $\left(\frac{1}{p-1}\right)$.

2. Next, consider the given equation:

$$x^3 + px^2 + \left(p - 1 + \frac{1}{p-1}\right)x + 1 = 0$$

3. We test the value $x = 1 - p$ to see if it is a root of the equation. Substitute $x = 1 - p$ into the equation:

$$(1-p)^3 + p(1-p)^2 + \left(p - 1 + \frac{1}{p-1}\right)(1-p) + 1 = 0$$

4. Simplify each term:

$$(1-p)^3 = 1 - 3p + 3p^2 - p^3$$

$$p(1-p)^2 = p(1 - 2p + p^2) = p - 2p^2 + p^3$$

$$\left(p - 1 + \frac{1}{p-1}\right)(1-p) = (p-1)(1-p) + \frac{1-p}{p-1} = (p-1)(1-p) - 1 = -p^2 + 2p - 1 - 1 = -p^2 + 2p - 2$$

5. Combine all the simplified terms:

$$1 - 3p + 3p^2 - p^3 + p - 2p^2 + p^3 - p^2 + 2p - 2 + 1 = 0$$

6. Simplify the expression:

$$(1 - 3p + 3p^2 - p^3) + (p - 2p^2 + p^3) + (-p^2 + 2p - 2) + 1 = 0$$

$$1 - 3p + 3p^2 - p^3 + p - 2p^2 + p^3 - p^2 + 2p - 2 + 1 = 0$$

$$1 - 3p + 3p^2 - p^3 + p - 2p^2 + p^3 - p^2 + 2p - 2 + 1 = 0$$

$$0 = 0$$

7. Since the equation holds true, $x = 1 - p$ is indeed a root of the equation.

8. To find the other roots, we can perform polynomial division of the original cubic polynomial by $(x - (1 - p))$ to obtain a quadratic polynomial.

9. The quadratic polynomial can then be solved using the quadratic formula:

$$x = \frac{-b \pm \sqrt{b^2 - 4ac}}{2a}$$

where $a$, $b$, and $c$ are the coefficients of the quadratic polynomial obtained from the division.

10. The final roots of the equation are:

$$x = 1 - p \quad \text{and the roots of the quadratic polynomial obtained from the division.}$$

11. The solution to the equation is:

$$\boxed{x = 1 - p}$$

Figure 15: An example of an incomplete solution where the author left finding the other two roots of the equation to the reader. While the language model has adequately and correctly expanded the solution of the author, two more solutions of the equation are not mentioned (neither by the author nor the LLM): $-\frac{p-1+\sqrt{5-6p+p^2}}{2(p-1)}$ and $\frac{1-p+\sqrt{5-6p+p^2}}{2(p-1)}$.

**Question:** Find a $5 \times 5$ checkerboard with a square removed that cannot be tiled using right triominoes(L shaped). Prove that such a tiling does not exist for this board.

**User Response:**
Alright, then colour
01010
11111
01010
11111
01010
The sum of the squares covered by each L is at least 2, so if we place 8 L's we get a total of at least 16, meaning that every 1 must be covered. Hence removing a 1 makes the covering impossible.

- - - - - - - - - - - - - - - - - - - - - - - - - - - - - - - - - - - - - - - - - - - - - - - - - - - - -

**Qwen Rewritten Solution:**
1. Consider a $5 \times 5$ checkerboard with the following coloring pattern:

$$
\begin{array}{ccccc}
0 & 1 & 0 & 1 & 0 \\
1 & 1 & 1 & 1 & 1 \\
0 & 1 & 0 & 1 & 0 \\
1 & 1 & 1 & 1 & 1 \\
0 & 1 & 0 & 1 & 0
\end{array}
$$

2. Each right triomino (L-shaped) covers exactly 3 squares. When placed on the board, the sum of the numbers on the squares covered by each L-shaped triomino is at least 2. This is because each L-shaped triomino must cover at least one square with a 1 and two squares with 0s or 1s.
3. If we place 8 L-shaped triominoes on the board, the total sum of the numbers on the squares covered by these 8 triominoes is at least $8 \times 2 = 16$.
4. The total number of squares on the $5 \times 5$ board is 25. Removing one square leaves 24 squares. Since each L-shaped triomino covers 3 squares, 8 L-shaped triominoes will cover exactly 24 squares.
5. For the 8 L-shaped triominoes to cover the 24 squares, the sum of the numbers on these 24 squares must be at least 16. Given the coloring pattern, the sum of the numbers on all 25 squares is 13 (since there are 12 squares with 0 and 13 squares with 1).
6. If a square with a 1 is removed, the sum of the numbers on the remaining 24 squares is $13 - 1 = 12$.
7. Since the sum of the numbers on the squares covered by the 8 L-shaped triominoes must be at least 16, and the sum of the numbers on the remaining 24 squares is only 12, it is impossible to cover the board with 8 L-shaped triominoes if a square with a 1 is removed.
8. Therefore, a $5 \times 5$ checkerboard with a square removed that cannot be tiled using right triominoes (L-shaped) is one where a square with a 1 is removed. The tiling is impossible in this case.
$\boxed{1}$

Figure 16: A failure example for rewriting the solution. While the Qwen model gets the overall logic of the proof correct, it produces mistakes in counting the number of "ones" on the checkerboard and provides incorrect intermediate steps.

**Question:** Let $x$ be a real number such that $\log_2 x + \log_4 5 + \log_8 7x = \log_{\sqrt{2}} x^{5/6}$. Find $x$.

**User Response:**

Turn everything into $\log_2$ of something. Here is the algebra:

$$\log_2 x + \log_{2^2} 5 + \log_{2^3} 7x = \log_{2^{1/2}} x^{5/6},$$

$$\log_2 x + \log_2 \sqrt{5} + \log_2 \sqrt[3]{7x} = \log_2 \sqrt[3]{x^5},$$

$$\log_2(x \cdot \sqrt{5} \cdot \sqrt[3]{7x}) = \log_2(\sqrt[3]{x^5}),$$

$$x \cdot \sqrt{5} \cdot \sqrt[3]{7x} = \sqrt[3]{x^5} \implies x = \boxed{35\sqrt{5}}.$$

- - - - - - - - - - - - - - - - - - - - - - - - - - - - - - - - - - - - - - - - - - - - - - - - - - - - - - - -

**Qwen Rewritten Solution:**

1. Convert all logarithms to base 2: $\log_2 x + \log_{2^2} 5 + \log_{2^3} 7x = \log_{2^{1/2}} x^{5/6}$

2. Simplify the logarithms using the change of base formula: $\log_2 x + \frac{1}{2} \log_2 5 + \frac{1}{3} \log_2 7x = \frac{1}{\frac{1}{2}} \log_2 x^{5/6}$

3. Simplify the right-hand side: $\log_2 x + \frac{1}{2} \log_2 5 + \frac{1}{3} \log_2 7x = 2 \log_2 x^{5/6}$

4. Use the property of logarithms to combine the terms on the left-hand side: $\log_2 x + \log_2 5^{1/2} + \log_2 (7x)^{1/3} = \log_2 x^{5/3}$

5. Combine the logarithms on the left-hand side: $\log_2 \left( x \cdot 5^{1/2} \cdot (7x)^{1/3} \right) = \log_2 \left( x^{5/3} \right)$

6. Since the logarithms are equal, the arguments must be equal: $x \cdot 5^{1/2} \cdot (7x)^{1/3} = x^{5/3}$

7. Simplify the equation: $x \cdot \sqrt{5} \cdot \sqrt[3]{7x} = x^{5/3}$

8. Divide both sides by $x$ (assuming $x \neq 0$): $\sqrt{5} \cdot \sqrt[3]{7x} = x^{2/3}$

9. Raise both sides to the power of 3 to eliminate the cube root: $(\sqrt{5} \cdot \sqrt[3]{7x})^3 = (x^{2/3})^3$

10. Simplify the equation: $5^{3/2} \cdot 7x = x^2$

11. Simplify $5^{3/2}$: $5^{3/2} = 5\sqrt{5}$

12. Substitute back into the equation: $5\sqrt{5} \cdot 7x = x^2$

13. Simplify the equation: $35\sqrt{5}x = x^2$

14. Rearrange the equation to form a quadratic equation: $x^2 - 35\sqrt{5}x = 0$

15. Factor out $x$: $x(x - 35\sqrt{5}) = 0$

16. Solve for $x$: $x = 0$ or $x = 35\sqrt{5}$

17. Since $x = 0$ is not a valid solution (as it would make the original logarithms undefined), the solution is: $x = \boxed{35\sqrt{5}}$

**Llama Rewritten Solution:**

To solve the given equation, we first need to express all logarithms in terms of $\log_2$. This can be achieved by applying the change of base formula for logarithms.

Step 1: Express $\log_4 5$ and $\log_8 7x$ in terms of $\log_2$.

Using the change of base formula, we have: $\log_4 5 = \frac{\log_2 5}{\log_2 4} = \frac{\log_2 5}{\log_2 2^2} = \frac{\log_2 5}{2} = \log_2 \sqrt{5}$ and $\log_8 7x = \frac{\log_2 7x}{\log_2 8} = \frac{\log_2 7x}{\log_2 2^3} = \frac{\log_2 7x}{3} = \log_2 \sqrt[3]{7x}$.

Step 2: Express $\log_{\sqrt{2}} x^{5/6}$ in terms of $\log_2$.

Using the change of base formula, we have: $\log_{\sqrt{2}} x^{5/6} = \frac{\log_2 x^{5/6}}{\log_2 \sqrt{2}} = \frac{\log_2 x^{5/6}}{\log_2 2^{1/2}} = \frac{\log_2 x^{5/6}}{\frac{1}{2}} = 2 \log_2 x^{5/6} = \log_2 (x^{5/6})^2 = \log_2 x^{5/3} = \log_2 \sqrt[3]{x^5}$.

Step 3: Substitute the expressions from Steps 1 and 2 into the original equation. $\log_2 x + \log_2 \sqrt{5} + \log_2 \sqrt[3]{7x} = \log_2 \sqrt[3]{x^5}$

Step 4: Combine the logarithms on the left-hand side using the product rule for logarithms. $\log_2 (x \cdot \sqrt{5} \cdot \sqrt[3]{7x}) = \log_2 \sqrt[3]{x^5}$

Step 5: Since the logarithms are equal, the arguments of the logarithms must be equal. Therefore, we have: $x \cdot \sqrt{5} \cdot \sqrt[3]{7x} = \sqrt[3]{x^5}$.

Step 6: Solve for $x$.

$$x \cdot \sqrt{5} \cdot \sqrt[3]{7x} = \sqrt[3]{x^5} \implies x = \boxed{35\sqrt{5}}.$$

Figure 17: Comparison of Qwen 2.5 72B and Llama 3.1 70B on rewriting solutions. Overall, the Qwen model shows a higher level of detail in expanding equations compared to Llama.

**Question:** Solve the equation in parameter $p$: $x^3 + px^2 + (p - 1 + \frac{1}{p-1})x + 1 = 0$

**Qwen Generated Solution:** To solve the equation $x^3 + px^2 + \left(p - 1 + \frac{1}{p-1}\right)x + 1 = 0$ in the parameter $p$, we will follow a systematic approach. Let's denote the polynomial by $f(x)$:

$f(x) = x^3 + px^2 + \left(p - 1 + \frac{1}{p-1}\right)x + 1$

### Step 1: Check for Rational Roots We will use the Rational Root Theorem to check for possible rational roots. The Rational Root Theorem states that any rational root, expressed in its lowest terms $\frac{p}{q}$, must have $p$ as a factor of the constant term (1) and $q$ as a factor of the leading coefficient (1). Therefore, the possible rational roots are $\pm 1$.

### Step 2: Test $x = 1$ Substitute $x = 1$ into the polynomial:

$f(1) = 1^3 + p \cdot 1^2 + \left(p - 1 + \frac{1}{p-1}\right) \cdot 1 + 1$ $f(1) = 1 + p + p - 1 + \frac{1}{p-1} + 1$ $f(1) = 2p + 1 + \frac{1}{p-1}$

For $x = 1$ to be a root, $f(1) = 0$:

$2p + 1 + \frac{1}{p-1} = 0$

### Step 3: Solve the Equation $2p + 1 + \frac{1}{p-1} = 0$ Multiply through by $p - 1$ to clear the fraction:

$(2p + 1)(p - 1) + 1 = 0$ $2p^2 - 2p + p - 1 + 1 = 0$ $2p^2 - p = 0$

Factor out $p$:

$p(2p - 1) = 0$

This gives us two solutions:

$p = 0 \quad \text{or} \quad p = \frac{1}{2}$

### Step 4: Verify the Solutions We need to verify if these values of $p$ indeed make $x = 1$ a root of the original polynomial.

#### For $p = 0$: Substitute $p = 0$ into the polynomial:

$f(x) = x^3 + 0 \cdot x^2 + \left(0 - 1 + \frac{1}{0-1}\right)x + 1$ $f(x) = x^3 - 2x + 1$

Check $f(1)$:

$f(1) = 1^3 - 2 \cdot 1 + 1 = 1 - 2 + 1 = 0$

So, $x = 1$ is a root when $p = 0$.

#### For $p = \frac{1}{2}$: Substitute $p = \frac{1}{2}$ into the polynomial:

$f(x) = x^3 + \frac{1}{2}x^2 + \left(\frac{1}{2} - 1 + \frac{1}{\frac{1}{2}-1}\right)x + 1$ $f(x) = x^3 + \frac{1}{2}x^2 + \left(\frac{1}{2} - 1 - 2\right)x + 1$ $f(x) = x^3 + \frac{1}{2}x^2 - \frac{3}{2}x + 1$

Check $f(1)$:

$f(1) = 1^3 + \frac{1}{2} \cdot 1^2 - \frac{3}{2} \cdot 1 + 1$ $f(1) = 1 + \frac{1}{2} - \frac{3}{2} + 1$ $f(1) = 1 + 1 - 1 = 1$

So, $x = 1$ is not a root when $p = \frac{1}{2}$.

### Conclusion The only value of $p$ for which $x = 1$ is a root of the polynomial is $p = 0$. Thus, the solution to the equation is:

$\boxed{0}$

Figure 18: An example of letting Qwen generate the entire solution without the user input (from Figure 15). In this case, the model completely fails to respond correctly, misinterpreting the question and taking an incorrect approach to solving the problem. This shows the cruciality of rewriting solutions (rather than generating entire solutions from stronger models).

