# OpenReview forum: "AoPS Dataset: Leveraging Online Olympiad-Level Math Problems for LLMs Training and Contamination-Resistant Evaluation"
_ICLR.cc/2025/Conference — Submitted to ICLR 2025_

### Official Review · Reviewer_vxhJ · 2024-10-29

**Soundness:** 1
**Presentation:** 2
**Contribution:** 1
**Rating:** 3
**Confidence:** 5

**Summary:**

In this paper, the authors prepared an instruct tuning dataset and a benchmark by collecting the math questions from a forum. The key steps of the data curation procedure are mostly clearly described. They also conducted experiments for finetuning a few open-source LLMs.

Overall, it is a reasonable work and might be able to provide the community with a useful resource. The technical contribution is not significant because it is more like a software system for data curation.

Moreover, the authors did not confirm what data they would release. As the major contribution of the paper, if the prepared datasets (instruct tuning and benchmark) will not be released, their contribution to the community will be significantly undermined.

**Strengths:**

In this paper, the authors prepared an instruct tuning dataset and a benchmark by collecting the math questions from a forum. The key steps of the data curation procedure are mostly clearly described. They also conducted experiments for finetuning a few open-source LLMs.

**Weaknesses:**

Overall, it is a reasonable work and might be able to provide the community with a useful resource. The technical contribution is not significant because it is more like a software system for data curation.

Moreover, the authors did not confirm what data they would release. As the major contribution of the paper, if the prepared datasets (instruct tuning and benchmark) will not be released, their contribution to the community will be significantly undermined.

Some other points:
- There is no discussion regarding the answer correctness in the instruct tuning data.
- For decontamination, I think 10-gram or 8-gram is sort of not enough. In Zhuo et al., 2024, 10-gram is used on coding data, but here is math data. They are quite different.

**Questions:**

1. In both training and benchmark, Qwen LLMs are used for rewriting, will this bring in bias?

2. When preparing answers for the testing questions, what’s the breakdown of each step? the boxed answer, and the agreed answers by the rewriting LLMs.

---

### Official Review · Reviewer_pSEP · 2024-11-03

**Soundness:** 3
**Presentation:** 3
**Contribution:** 2
**Rating:** 6
**Confidence:** 4

**Summary:**

This work presents a new olympiad-level math problem dataset with two main features: (1) a training set for instruction fine-tuning in math problem solving (AoPS-Instruct), and (2) an evaluation dataset creation pipeline that can periodically update test examples (LiveAoPSBench). The manuscript provides a clear and detailed description of the dataset creation process, including steps taken for data contamination prevention and quality control. The experiments conducted demonstrate the effectiveness of AoPS-Instruct in fine-tuning relatively small LLMs and highlight the characteristics of LiveAoPSBench.

**Strengths:**

1. The evolving nature of the proposed automatic pipeline, LiveAoPSBench, is important as it reduces the risk of data contamination.

2. The dataset creation process is clearly documented, with specific steps taken for quality control. Experimental results also demonstrate the effectiveness of the training set (in terms of its success in fine-tuning small LLMs) and the quality of the test set (as indicated by a high correlation with existing benchmarks).

3. The finding that benchmarked LLM performance declines on more recent test examples highlights the need for periodically updated benchmarks.

**Weaknesses:**

The dataset creation process relies heavily on LLMs, which may undermine the reliability and usefulness of the proposed datasets and evaluation pipeline, given that LLMs are not always dependable. More specifically,

1. The QA pairs are extracted by Llama3.1-70B-Instruct. As discussed in Section 4.4, Evaluation Quality Assessment, human annotators found 8% of the annotations to be incorrect and 4% to fall under the no-answer category, resulting in a combined error rate of 12%. Such noise can be problematic, especially when evaluating state-of-the-art models whose error rates may not be significantly higher. Additionally, a 91% agreement rate between human annotators is reported. While it is understandable that olympiad-level problems are challenging, as explained in the manuscript, this still indicates a degree of unreliability in human evaluation itself, as mathematical problems should ideally have objective correctness. This suggests that the actual error rate of the dataset construction pipeline might be higher, considering the noise in human annotation.

2. The step-by-step solutions in the training set are rewritten by Qwen 2.5 72B, so it is likely that the performance of models fine-tuned on this training set will be limited by the performance of Qwen 2.5 72B. This makes the training set more suitable for use in a distillation setting to fine-tune smaller models. However, it might be less effective for training larger models. Notably, the training experiments conducted are also on much smaller models, which aligns with a distillation approach.

**Questions:**

Regarding the "Weaknesses" section, what caused disagreement among human annotators when verifying answer correctness? Was it due to (1) mistakes by some annotators, (2) uncertainty among annotators, or (3) inherent subjectivity in assessing correctness? (These factors might also have contributed together).

---

### Official Review · Reviewer_Qqyj · 2024-11-04

**Soundness:** 4
**Presentation:** 4
**Contribution:** 3
**Rating:** 8
**Confidence:** 4

**Summary:**

Owing to the structured nature of math problem solving that requires not just recall of facts but also, understanding, abstraction and reasoning, it is one of the most challenging tasks for LLMs. While there are existing math datasets like, GSM8K and MATH, they have reached a level of saturation with SOTA models and are now susceptible to contamination. Newer datasets, like, OlympiadBench and OmniMath temporarily mitigate the above problems but remain susceptible to these issues as LLMs continue to evolve. Moreover, creation of these datasets, especially complex Olympiad-level problems, at scale is time and cost intensive. Motivated by these challenges, the authors argue for the need for scalable and automated methods to collect high-quality data for Olympiad-level problems to facilitate further advancements in this field. It is also crucial that these evaluation benchmarks be evolving and contain abundant and up-to-date test samples.
Towards that the authors leverage the raw AoPS forum data and propose a pipeline to extract questions and solutions leading to AoPS-Instruct, a novel large-scale dataset with 666.1K Olympiad-level math QA pairs. Using the most recent QA pairs, they develop an automatic pipeline that introduces LiveAoPSBench, a contamination-resistant evaluation set. Their experiments on LiveAoPSBench show a declining performance trend over time for various LLMs that improves after SFT on AoPS-Instruct.

**Strengths:**

As LLMs continue to progressively evolve, there is a growing need for more complex evaluation benchmarks that challenge the capabilities of these models. The evaluation also needs to be trustworthy and the onus lies equally on those training models as well as those publishing evaluation benchmarks. I think this paper does justice to both these requirements and in that it attempts to address an important aspect of the LLM research and development.
In doing so, the authors take into account the effort and cost involved in building such benchmarks and propose an automated pipeline that is able to produce complex Olympiad-level QA pairs at scale. To the best of my knowledge, the ideas presented here are original. Although the approach leverages community QA data from AoPS forums, the steps involved in curating QA pairs from the raw data are non-trivial.
The writing of the paper is clear and the ideas and the methodology are well presented. The authors also conduct thorough evaluation to justify the complexity and trustworthiness of their benchmark datasets.

**Weaknesses:**

Their evaluation dataset currently focuses on boxed answers and excludes proof questions. Although the authors highlight this as their current limitation, I think proof questions form a significant part of the Olympiad-level questions. Excluding them might considerably limit the scope of evaluation of LLMs for their reasoning abilities.
Expanding the scope to proof questions would require a thought around evaluation as well. While boxed answers are more amenable to objective evaluation, proof questions might require a more subjective evaluation, potentially leveraging LLMs as a judge.
I also found no references to LLM as judge as striking. While it might not be a necessary tool considering the current scope, I think my meta point is that the authors should have touched upon these aspects and shown some evidence of early work / experiments. That would further strengthen the contributions of this work and its application to further the state-of-the-art in LLM research.

**Questions:**

Results in Table 2 for AIME24 and AMC23 are inconsistent. Why are certain models able to perform better on one vs the other?

---

### Official Review · Reviewer_sDbk · 2024-11-04

**Soundness:** 4
**Presentation:** 4
**Contribution:** 4
**Rating:** 8
**Confidence:** 4

**Summary:**

This paper created a dataset in one of the important areas of mathematical reasoning, IMO problem solving, where the LLMs currently suffer. Such a open-source dataset with 650K samples and benchmark can be a

1. They presented the dataset creation pipeline, comes with quality filtering, solution rewriting.
2. Also they also presented the automated pipeline for liveAoPSbench for evaluating recent LLM to avoid contamination.
3. The conduct experiments with SFT over open-source models to demonstrate the presented dataset.

**Strengths:**

1. Dataset can be a very important contribution to the community, both training and LiveBench are ready.
2. The pipeline can be used for other domain/scenarios to create something similar and essential for current LLM.
3. The instruction-tuning performance clearly show that the dataset is useful for improving the performance on IMO.

**Weaknesses:**

1. It’s again very difficult to guarantee the quality of the data. How are we going to update the data performance time by time? Otherwise, the bad data will still affect the performance anyway.

**Questions:**

As mentioned in the weakness.

---

### Meta-Review · Area_Chair_Gmqg · 2024-12-22

**Metareview:**

The paper introduces two contributions to mathematical reasoning for LLMs:
*AoPS-Instruct* is a dataset of 650,000 Olympiad-level math question-answer pairs derived from the Art of Problem Solving (AoPS) forum, designed for instruction tuning.
*LiveAoPSBench*: An evaluation benchmark that updates over time, highlighting declining LLM performance on newer problems.
The paper shows that fine-tuning LLMs on AoPS-Instruct improves reasoning benchmarks and provides insights into pretraining contamination effects.

Strengths:
- Practical relevance: Several reviewers (sDbk, Qqyj, pSEP) highlight the importance of scalable, high-quality datasets and benchmarks for mathematical reasoning, emphasizing the utility of LiveAoPSBench in reducing contamination and providing up-to-date evaluation.
- Focus on contamination: Reviewers acknowledged the importance of benchmarks in addressing contamination, which LiveAoPSBench aims to do by dynamically updating the benchmark
- Extensive Experiments: Reviewers (sDbk, pSEP) commend the extensive evaluation and performance improvements shown with AoPS-Instruct.

Weaknesses:
- Methodological novelty: The dataset creation process primarily uses existing LLMs for extracting and rewriting forum content (vxhJ).
- Data Quality Issues: Reviewers (vxhJ, pSEP) raise concerns about the reliability of the datasets, citing a notable error rate in benchmark annotations and the dependence on rewriting models for training data.
- Limited Benchmark Scope: Reviewer Qqyj notes that LiveAoPSBench excludes proof-based questions, which are crucial for fully evaluating mathematical reasoning capabilities.

Overall, the practical contributions and insights into dataset contamination and model degradation are acknowledged by multiple reviewers. However, while during discussions some of the concerns were resolved, issues regarding quality of the dataset and the methodological rigor and novelty remain unresolved (even after internal discussion).

**Additional Comments On Reviewer Discussion:**

Data Quality: The authors addressed concerns raised by pSEP and vxhJ by refining prompts and analyzing errors, which improved human agreement from 88% to 91%. However, reviewers (pSEP) suggest further efforts to improve accuracy and perform detailed error analyses for reliability in mathematical tasks.
Exclusion of Proof Questions: The authors acknowledged this limitation raised by Qqyj but justified it as out of scope for the current work while expressing interest in future extensions.
Perceived Contribution: Reviewer vxhJ maintained that the paper’s primary contribution—dataset creation—lacked sufficient technical novelty. The authors argued that similar works, like Llemma, have been accepted to ICLR, and such works also target finetuning models and releasing math-specific datasets.
However, it could be argued that Llema had a significantly larger scope, by pretraining large-scale models, releasing a large-scale training data that was a collective effort of curation and synthetic generation, incorporating tool usage, as well as matching performance of the proprietary math model (Minerva), while providing many additional analysis.

---

### Decision · Program_Chairs · 2025-01-22

Reject